# Digital RNA sequencing using unique molecular identifiers enables ultrasensitive RNA mutation analysis
Manuel Luna Santamaría [1,2], Daniel Andersson [1], Toshima Z. Parris[3], Khalil Helou[3],
Tobias Österlund [1,2,4] & Anders Ståhlberg [1,2,4] ✉

Mutation analysis is typically performed at the DNA level since most technical approaches are developed for DNA analysis. However, some applications, like transcriptional mutagenesis, RNA editing and gene expression analysis, require RNA analysis. Here, we combine reverse transcription and digital DNA sequencing to enable low error digital RNA sequencing. We evaluate yield, reproducibility, dynamic range and error correction rate for seven different reverse transcription conditions using multiplexed assays. The yield, reproducibility and error rate vary substantially between the specific conditions, where the yield differs 9.9-fold between the best and worst performing condition. Next, we show that error rates similar to DNA sequencing can be achieved for RNA using appropriate reverse transcription conditions, enabling detection of mutant allele frequencies <0.1% at RNA level. We also detect mutations at both DNA and RNA levels in tumor tissue using a breast cancer panel. Finally, we demonstrate that digital RNA sequencing can be applied to liquid biopsies, analyzing cell-free gene transcripts. In conclusion, we demonstrate that digital RNA sequencing is suitable for ultrasensitive RNA mutation analysis, enabling several basic research and clinical applications.

The ability to analyze gene expression by sensitive and specific methods, such as quantitative PCR, digital PCR, microarrays and massive parallel sequencing, is fundamental in basic research as well as in clinical applications. Today, individual genes are typically assessed by PCR-based approaches, while transcriptomes are analyzed by sequencing. To enable RNA analysis, RNA needs to be reverse transcribed into complementary DNA (cDNA). Reverse transcription (RT) is a reproducible reaction for a given sequence, but the reaction efficiency is highly variable between different genes and sequences[1,2]. The RT efficiency also depends on the choice of reverse transcriptase, reaction conditions and priming strategy[2–7]. Several applications require the ability to detect low variant allele frequencies at RNA level. For instance, when the amount of sample is small it may be beneficial to perform mutation analysis at RNA level, since the number of transcripts is higher than the number of genomic DNA molecules for most expressed genes. Furthermore, combined DNA and RNA analysis may also provide superior sensitivity to detect mutations in blood plasma[8]. There are

also emerging applications that require analysis at RNA level. For example, errors that occur during cellular transcription, i.e., transcriptional mutagenesis, may be both biologically and clinically relevant, where variations in RNA sequences even at low frequencies may impact cellular functions and phenotypes[9–11]. Another area is the analysis of RNA modifications[12]. Post transcriptional RNA editing of adenosine to inosine by the ADAR protein family is a common RNA modification that is associated with diseases, including several tumor entities[12,13].

To improve the accuracy of sequencing, unique molecular identifiers (UMIs) can be attached to the cDNA molecules during or after RT[14]. The UMI usually consists of a random 6–16 nucleotide long sequence. Bioinformatically, reads with the same UMI are collapsed to a single consensus read, reducing quantification biases that arise when the same cDNA molecule is sequenced multiple times[15]. Polymerase-induced errors also accumulate during library construction and cluster generation during sequencing, limiting the ability to detect variant allele frequencies below

[1]Sahlgrenska Center for Cancer Research, Department of Laboratory Medicine, Institute of Biomedicine, Sahlgrenska Academy at University of Gothenburg, Gothenburg, Sweden. [2]Wallenberg Centre for Molecular and Translational Medicine, University of Gothenburg, Gothenburg, Sweden. [3]Sahlgrenska Center for Cancer Research, Department of Oncology, Institute of Clinical Sciences, Sahlgrenska Academy, University of Gothenburg, Gothenburg, Sweden. [4]Region Västra Götaland, Sahlgrenska University Hospital, Department of Clinical Genetics and Genomics, Gothenburg, Sweden. ✉e-mail: anders.stahlberg@gu.se

1–5%[16–18]. Unique molecular identifiers can also be utilized to overcome this issue for RNA analysis, but this concept is poorly studied. To enable correction of polymerase-induced errors, each initial molecule needs to be sequenced multiple times[19,20]. When trying to detect very low variant allele frequencies this means that large numbers of cDNA molecules need to be sequenced, i.e., deep sequencing. For example, detection of 0.1% variant allele frequency means that on average, sequence reads from 1000 unique cDNA molecules need to be observed with multiple reads per UMI in order to detect one true variant cDNA molecule. Thus, several thousand reads are required per target cDNA sequence. Hence, polymerase-induced error correction with UMIs is suited predominantly for targeted sequencing. For DNA analysis, several UMI-based strategies have been developed for targeted sequencing, such as SiMSen-Seq[21], CAPP-Seq[22], Duplex-Seq[23] and RareSeq[24]. However, the adaption and use of UMI-based targeted sequencing approaches for error-free cDNA sequencing is not developed. The properties of RT in digital RNA sequencing and mutation analysis are also mainly unknown.

Here, we report a digital RNA sequencing approach based on UMIs, originating from the SiMSen-Seq technology. We evaluated the properties of seven different RT approaches, determining cDNA yield, reproducibility and error rates. We also studied the dynamic range analyzing different RNA concentrations and the ability to detect mutations for one well-performing RT condition. For comparison, we analyzed identical sequences at both the DNA and RNA level. Next, we demonstrated the use of digital RNA sequencing to assess circulating mRNA extracted from blood plasma. The development of digital RNA sequencing enables several basic research and clinical applications that require accurate quantification and low error RNA sequencing.

## Results

### Experimental design to analyze RNA and reverse transcription properties using digital sequencing

Figure 1 shows the experimental workflows for digital RNA and DNA sequencing using SiMSen-Seq. For digital RNA sequencing, RNA first needs to be reverse transcribed into cDNA. SiMSen-Seq consists of two rounds of PCR. In the first barcoding PCR, target cDNA molecules are tagged with UMIs. In the second adapter PCR, sequencing adapters are attached to all targeted DNA. Libraries are finally purified and sequenced. This approach enables digital sequencing, correcting for both polymerase-induced errors and quantification biases[25] (Supplementary Fig. 1). The difference when analyzing cDNA compared with genomic DNA is that single-stranded cDNA generates three UMIs per original cDNA molecule, while double-stranded DNA produces six UMIs per molecule (Fig. 1). A schematic overview of UMI generation during barcoding PCR is shown in

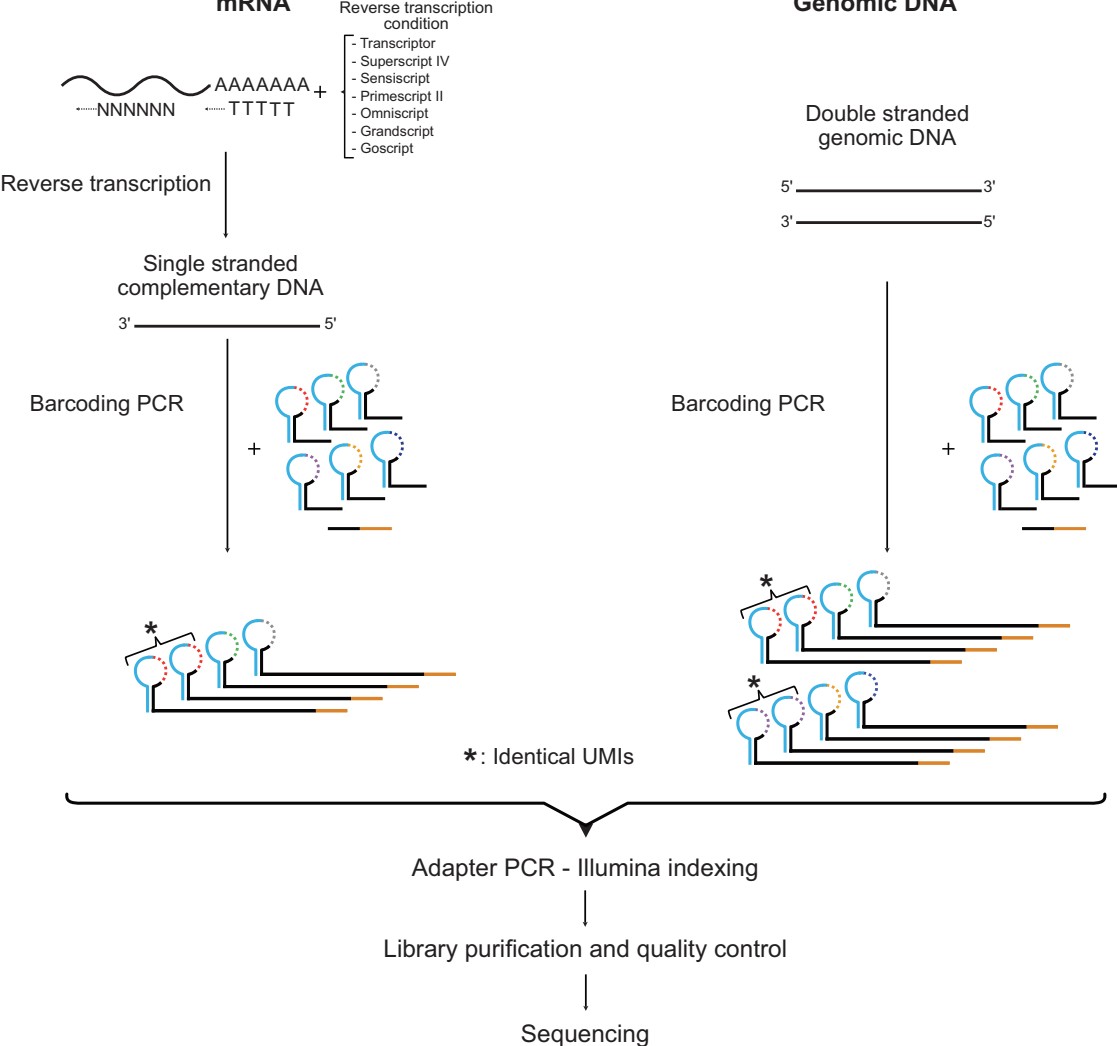

**Fig. 1 | Schematic overview of SiMSen-Seq.** The SiMSen-Seq workflow for RNA and DNA analysis, respectively. Complementary DNA is single-stranded, while genomic DNA is double-stranded, resulting in three and six different UMIs, respectively. One-third of the reaction volume of barcoding PCR is loaded into the adapter PCR. The tested RT conditions are shown.

**Table 1 | Reverse transcription conditions**

| Name (Company) | Reverse transcriptase origin[a] | RNase activity | Priming strategy[a] | Reaction volume (µL) | Reagents[a] | Temperature profile[b] |
|---|---|---|---|---|---|---|
| Transcriptor (Roche) | Recombinant enzyme in *E. coli* (10 U) | RNaseH+ | Oligo-dT (2.5 µM) Rand hex (2.5 µM) | 20 | RT buffer (1x) MgCl$_2$ (8 mM) dNTPs mix (1 mM) RNase inhibitor (20 U) | 65 °C inc. (10 min) 25 °C (5 min) 55 °C (30 min) 85 °C (5 min) hold at 4 °C |
| Superscript IV (Thermo-Fisher Scientific) | *MMLV* (200 U) | RNaseH+ | Oligo-dT (2.5 µM) Rand hex (2.5 µM) | 20 | RT buffer (1x) dNTPs mix (0.5 mM) DTT (5 nM) RNaseOUT (40 U) | 65 °C inc. (5 min) 23 °C (10 min) 50 °C (10 min) 80 °C (10 min) hold at 4 °C |
| Sensiscript (Qiagen) | Recombinant enzyme in *E. coli* (20U) | RNaseH + | Oligo-dT (1 µM) Rand hex (1 µM) | 10 | RT buffer (1x) dNTPs mix (0.5 mM) RNaseOUT (20 U) | 37 °C (60 min) hold at 4 °C |
| PrimeScript II (Takara) | *MMLV* (200 U) | n.p. | Oligo-dT (2.5 µM) Rand hex (2.5 µM) | 20 | RT buffer (1x) dNTPs mix (0.5 mM) RNase inhibitor (40 U) | 65 °C inc. (5 min) 30 °C (10 min) 42 °C (45 min) 95 °C (5 min) hold at 4 °C |
| Omniscript (Qiagen) | Recombinant enzyme in *E. coli* (20 U) | RNaseH + | Oligo-dT (1 µM) Rand hex (1 µM) | 10 | RT buffer (1x) dNTPs mix (0.5 mM) RNaseOUT (20 U) | 37 °C (60 min) hold at 4 °C |
| Grandscript (TATAA Biocenter) | *MMLV* | n.p. | Oligo-dT Rand hex | 10 | RT buffer (1x) MgCl$_2$ dNTPs Stabilizers | 22 °C inc. (5 min) 42 °C (60 min) 85 °C (5 min) hold at 4 °C |
| Goscript (Promega) | *MMLV* (80 U) | n.p. | Oligo-dT (5 µM) Rand hex (12 µM) | 20 | RT buffer (1x) MgCl$_2$ (1.875 mM) dNTPs mix (0.5 mM) RNase inhibitor (20 U) | 70 °C inc. (5 min) 25 °C (5 min) 42 °C (45 min) 70 °C (15 min) hold at 4 °C |

[a]Final reaction concentrations are shown. If not indicated, the information is not provided by the manufacturer.
[b]Initial preincubation time (Inc.) without reverse transcriptase is indicated.
n.p., information not provided by the manufacturer; *MMLV*, moloney murine leukemia virus; Rand hex, random hexamers.

Supplementary Fig. 2. After barcoding PCR, each original cDNA and genomic DNA molecule generates about one and two reads with specific UMIs into the adapter PCR, respectively, due to the three-fold dilution step between the two PCR reactions. We required at least three raw reads per UMI to generate consensus reads. An overview of the bioinformatics pipeline is shown in Supplementary Fig. 3.

We analyzed the performance of seven different RT conditions by analyzing five different sequence regions in the tumor protein p53 (*TP53*) gene (Supplementary Fig. 4). Total RNA was extracted from the myxoid liposarcoma cell line MLS 1765–92, where the same batch of RNA was used for all experiments that were compared. Reverse transcription was performed according to the manufacturers' instructions and Table 1. The seven RT conditions used different reverse transcriptase, reagents composition including their concentrations. The same assays were also used to analyze genomic DNA extracted from the MLS 1765–92 cells as reference. Figure 2 shows typical sequencing data before and after error correction using UMIs when analyzing RNA with two different RT conditions, Transcriptor or Omniscript, as well as genomic DNA. The error at each nucleotide position was defined as the number of non-reference reads divided by the total number of detected reads. The error rate was reported as the mean across all analyzed nucleotide positions per assay or panel. The observed error rates are the sum of all errors that accumulate during the entire analysis, which also includes biological variations among molecules. The error rate of the RT step can only be indirectly assessed by comparing data between the RT conditions with genomic DNA that does not require any RT step.

**Complementary DNA yield is reverse transcription-dependent**
To study the RT properties we evaluated cDNA yield, reproducibility and error rate in each of the seven different RT conditions. Figure 3a shows the relative cDNA yields for all RT conditions. Transcriptor generated the

highest cDNA yield followed by Primescript II and Omniscript, while Superscript IV produced the lowest cDNA yield. The mean difference in cDNA yields between Transcriptor and Superscript IV was 9.9-fold and the trend was similar for all five individual *TP53* assays (Fig. 3b). We also analyzed 40 ng genomic DNA as reference, which corresponds to ~11,200 molecules[26] that is equivalent to 22,400 consensus reads. We detected 54% of all loaded genomic DNA molecules after sequencing and data analysis.

Next, we compared the experimental RT reproducibility followed by SiMSen-Seq comparing the variability as coefficient of variation between technical replicates using genomic DNA as control (Fig. 3c). The coefficients of variation ranged from 2.0% for Goscript to 69.9% for Superscript IV with a mean of 26.7% across all seven RT conditions, while we observed a coefficient of variation of 6.2% for genomic DNA analysis. The reproducibility was expected to be inferior for RNA analysis since the additional RT step is known to increase experimental variability[2,27]. Based on cDNA yield and reproducibility data we selected Omniscript as the optimal RT condition for downstream experiments.

To assess the dynamic range of digital RNA sequencing with Omniscript in the RT step we performed dilution series using 140–8.75 ng total RNA (Fig. 3d). We observed a linear correlation between observed cDNA molecules and loaded amount of RNA molecules for all five *TP53* assays. For all assays we observed a saturation effect in RT yields for the highest loaded RNA concentration, i.e., fewer molecules were detected than expected.

To determine the variability between genes and different amounts of total RNA, we designed a hotspot mutation panel with eight assays targeting commonly mutated nucleotide position in *BRAF*, *EGFR*, *FLT3*, *KRAS*, *MEK1*, *NOTCH1* and *NRAS*. Figure 3e shows the number of consensus reads using Omniscript and three different amounts of total RNA ranging from 1 ng–200 ng. *EGFR*, *KRAS*, *MEK1* and *NRAS* were highly expressed in MLS 1765–92 cells, while *BRAF* and *NOTCH1* were intermediately expressed and *FLT3* was lowly expressed.

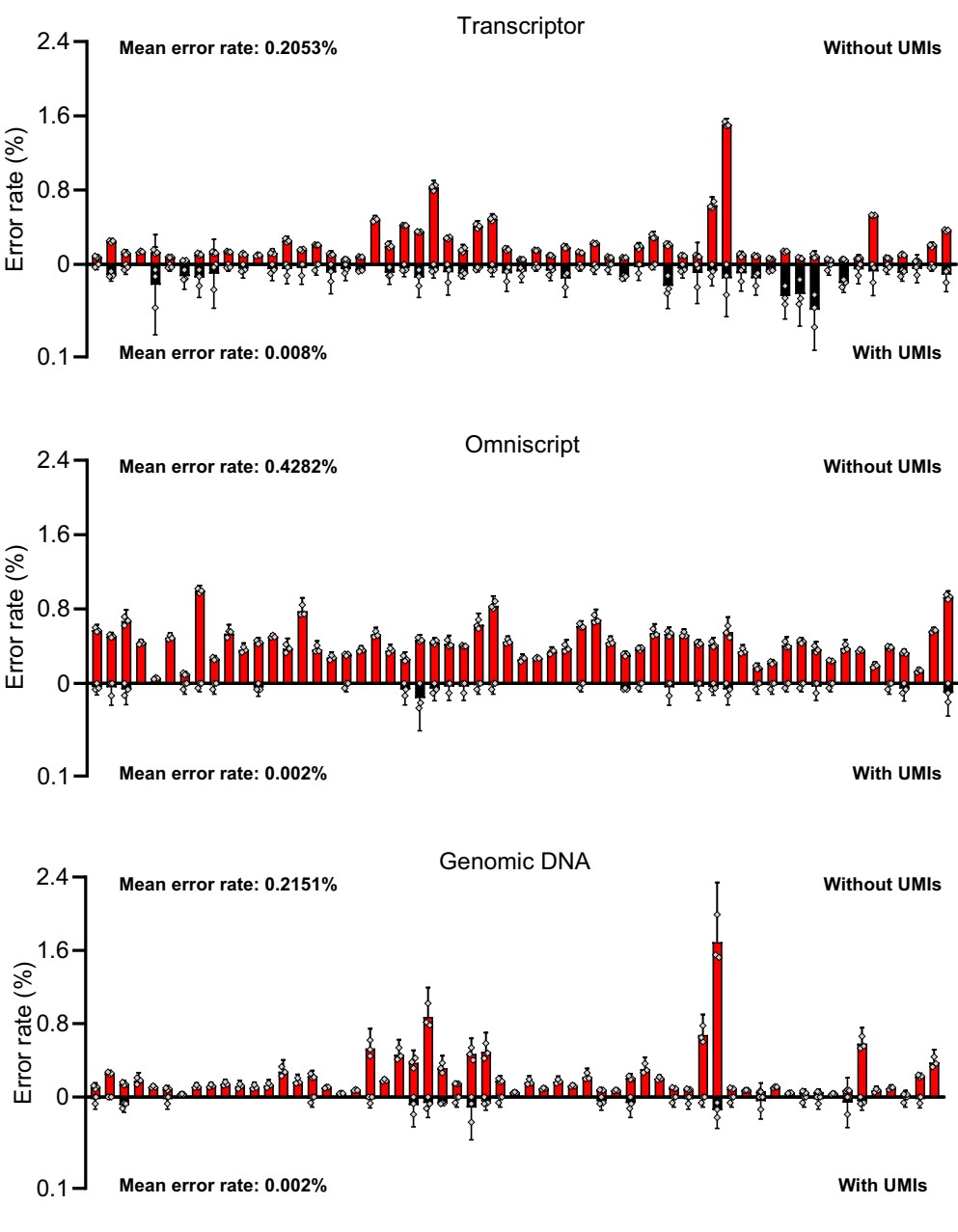

**Fig. 2 | Digital sequencing using UMIs.** Examples of sequencing data with and without error correction using UMIs. Total RNA and genomic DNA were analyzed from the same MLS 1765–92 cell line. Each position represents a specific nucleotide position in assay B, *TP53*. The error rate per position was calculated as the total number of non-reference reads divided by the total number of detected reads with and without UMI-error correction. Mean + 95% CI is shown, *n* = 3. The individual samples are not shown for nucleotide position with only zeros. The count matrices for these data with and without UMI-error correction is shown in Supplementary Data 1.

## Optimal reverse transcription condition enables digital RNA sequencing with low error rates

To enable ultrasensitive mutation detection at the RNA level, the technical sequencing error rate needs to be significantly lower than the desired variant allele frequency detection level. Figure 4a shows the mean error rates for the seven RT conditions and genomic DNA. Highest error rates were detected for Superscript IV followed by Transcriptor and Primescript II, while Sensiscript, Omniscript and Goscript generated the least errors. Figure 4b shows the error rates for the five individual *TP53* assays and Supplementary Fig. 5a, b display the error rates per nucleotide position with and without UMI-error correction. Superscript IV and Transcriptor produced the highest error rates for all assays, while Grandscript and Primescript II

generated variable error rates between the assays. Sensiscript, Omniscript and Goscript generated low error rates similar to the levels observed for genomic DNA analysis for all *TP53* assays. As expected, the number of non-reference molecules scaled with the total number of analyzed RNA molecules (Fig. 4c). To assess the benefits of using UMIs for error correction, we calculated the error correction factor, which is the ratio between the mean error rate with raw reads and consensus reads, for all RT conditions (Fig. 4d). The improvement using UMIs varied between 15.8 times for Transcriptor and 352.3 times for Sensiscript, while the error correction factor for genomic DNA was 122.7 times. The maximum mean error rates for the hotspot mutation panel was 0.0055% for *NOTCH1* using Omniscript (Fig. 4e), which were in the same range as for

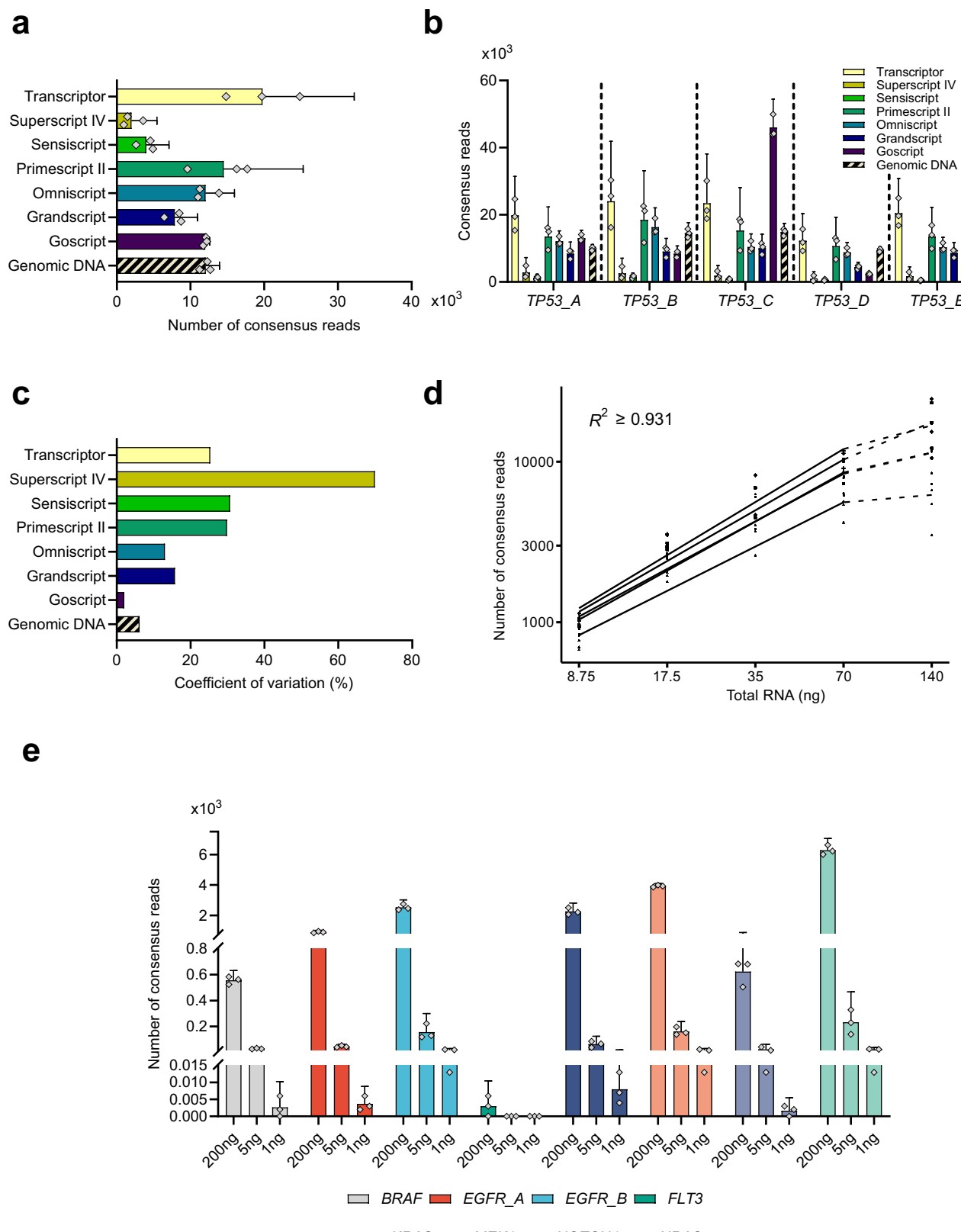

**Fig. 3 | Reverse transcription properties. a** Complementary DNA yield. For RNA analysis one consensus read corresponds to one cDNA molecule, while two consensus reads correspond to one genomic DNA molecule. The Sensiscript cDNA yields are scaled by a factor of four to compensate for lower amount of loaded RNA into RT and subsequent SiMSen-Seq. Mean + 95% CI is shown, $n = 3$. **b** Number of consensus reads for all individual *TP53* assays. Mean + 95% CI is shown, $n = 3$. **c** Coefficient of variation among individual RT conditions calculated using consensus reads. Note that replicates were performed at RNA level when evaluating RT conditions, $n = 3$. **d** Dynamic range of Omniscript. Dilution series ranging from 140–8.75 ng total RNA (Supplementary Data 2). One sample at 35 ng total RNA was considered outlier and removed. Linear regression was performed for each assay using the four lowest RNA concentrations to guide the eye, $n = 2$–3. **e** Complementary DNA yield for hotspot mutation panel. Omniscript was used with 200, 5 and 1 ng total RNA. Mean + 95% CI is shown, $n = 3$.

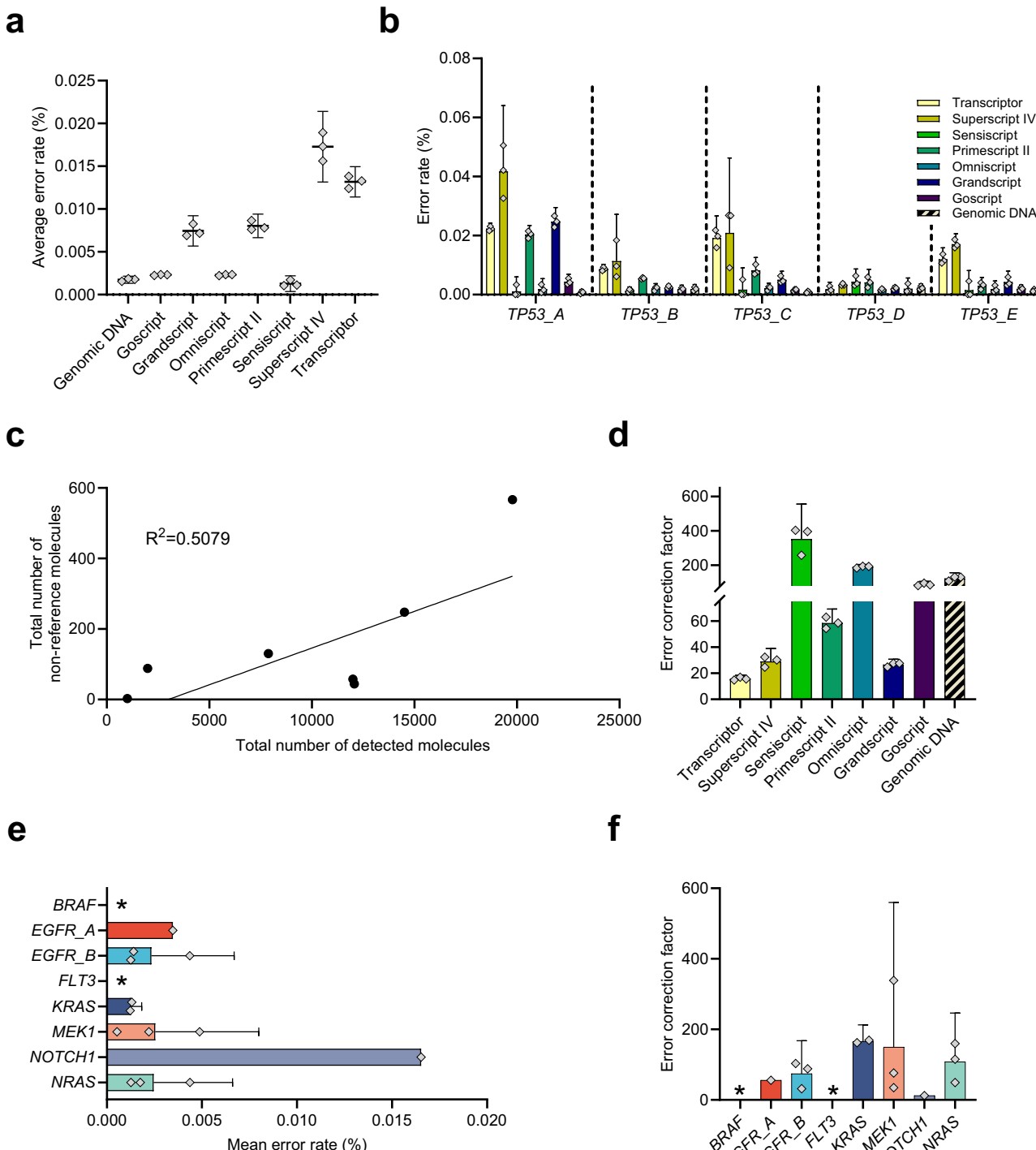

**Fig. 4 | Error rates in digital RNA sequencing. a** Mean error rates. The error rate was calculated as the total number of non-reference reads divided by the total number of detected reads per nucleotide position using consensus reads. The errors were then averaged for all nucleotide positions and assays. Mean + 95% CI is shown, *n* = 3. **b** Error rate per individual assay. Mean + 95% CI is shown, *n* = 3. **c** Linear relationship between total number of non-reference molecules compared with total number of detected molecules using consensus reads. The mean for each RT condition and genomic DNA is shown. The linear regression is shown. **d** Error correction factor using UMIs. The error rates before and after using UMIs were calculated and compared. Mean + 95% CI is shown, *n* = 3. **e** Error rates for individual assays in the hotspot mutation panel. Omniscript was used with 200 ng total RNA. Mean + 95% CI is shown, *n* = 3. * indicates no detected error. **f** Error correction factor using UMIs for hotspot mutation panel. The error rates before and after UMI correction were calculated and compared. Mean + 95% CI is shown, *n* = 3. * indicates that no factor value could be calculated since no errors were detected after UMI correction.

the *TP53* assays. The error correction factor ranged between 12.5 times for *NOTCH1* and 165.8 times for *KRAS* (Fig. 4f). For two assays we found no errors after UMI correction and could hence not calculate any correction factor. The same two assays were also the two lowest expressed

genes. We conclude that RT followed by SiMSen-Seq can reduce the technical mean error rates to <0.01% for the well-performing Omniscript, thus enabling the detection of mutations at <0.1% allele frequency, which is similar to DNA analysis.

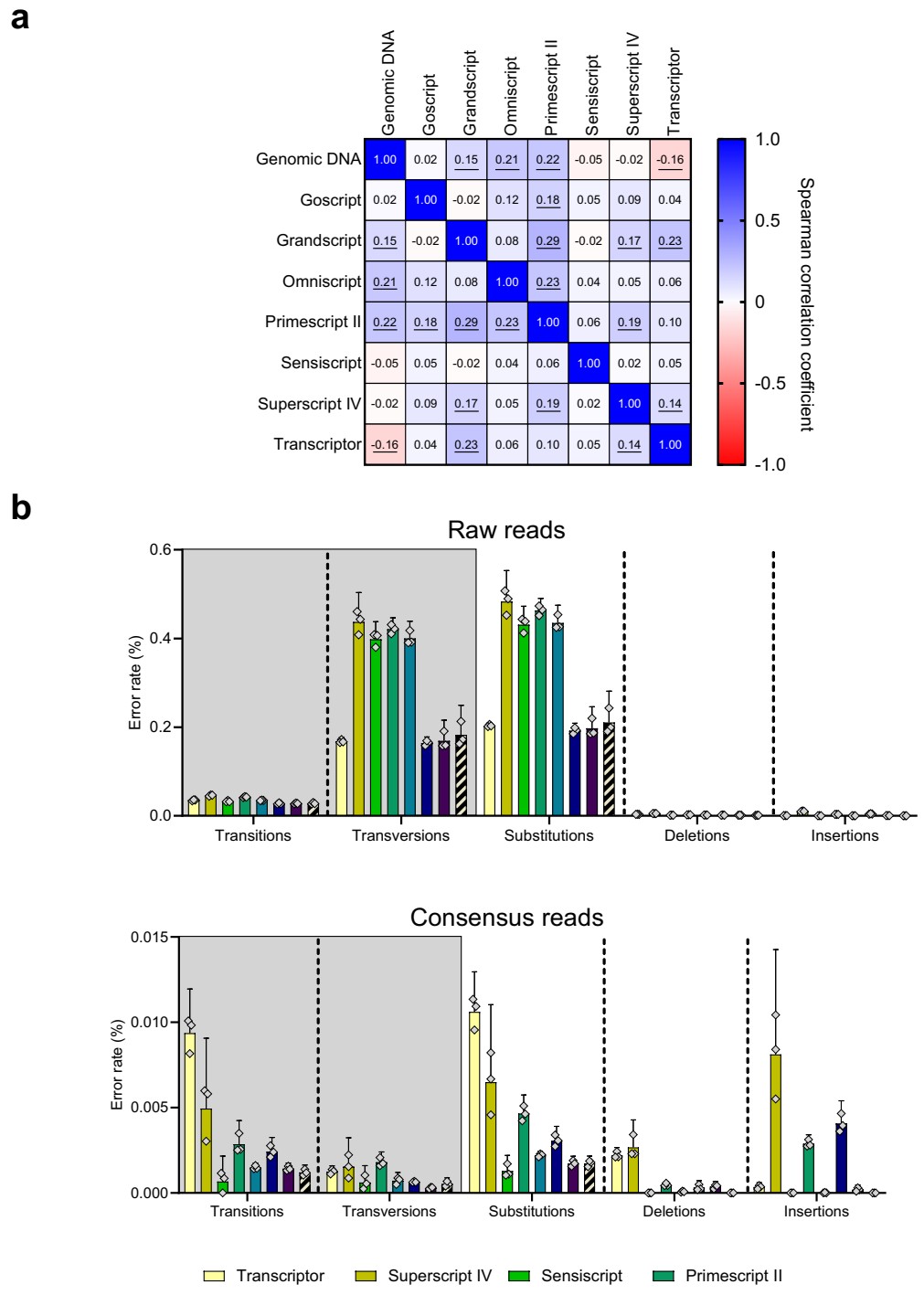

**Fig. 5 | Types of errors in digital RNA sequencing. a** Sequence-context dependent errors. Spearman correlation coefficients for error rates between all RT conditions and genomic DNA. The error rate was calculated for each nucleotide position using data for all five TP53 assays. Statistically significant values ($p < 0.05$) are underlined, $n = 3$. **b** Error rates for substitutions including transitions and transversions (gray background), deletions and insertions for all RT conditions and genomic DNA. Data are shown for raw reads and UMI-error corrected reads. Mean + 95% CI is shown, $n = 3$.

In general, we observed no or weak correlations between variant allele frequencies and nucleotide positions for most RT conditions as well as for genomic DNA, indicating that the sequence context influenced error rates only to a minor degree (Fig. 5a and Supplementary Fig. 6). Interestingly, we observed one nucleotide position that generated an error rate between 0.432% and 1.38% for Transcriptor, Superscript IV, Primescript II and Grandscript, while the error rates for the other RT conditions and genomic DNA was <0.075% (Supplementary Fig. 5b). Next, we compared the types of

errors for all RT conditions with genomic DNA as reference (Fig. 5b). Overall, we observed similar or higher amounts of substitutions, including transitions and transversions, for all RT conditions compared with genomic DNA. We observed deletions and insertions for all RT conditions except Sensiscript. However, Sensiscript also generated the lowest cDNA yield and we may not have had enough cDNA molecules to allow the detection of rare deletions and insertions. For genomic DNA analysis, we detected no deletions and insertions in the *TP53* assays.

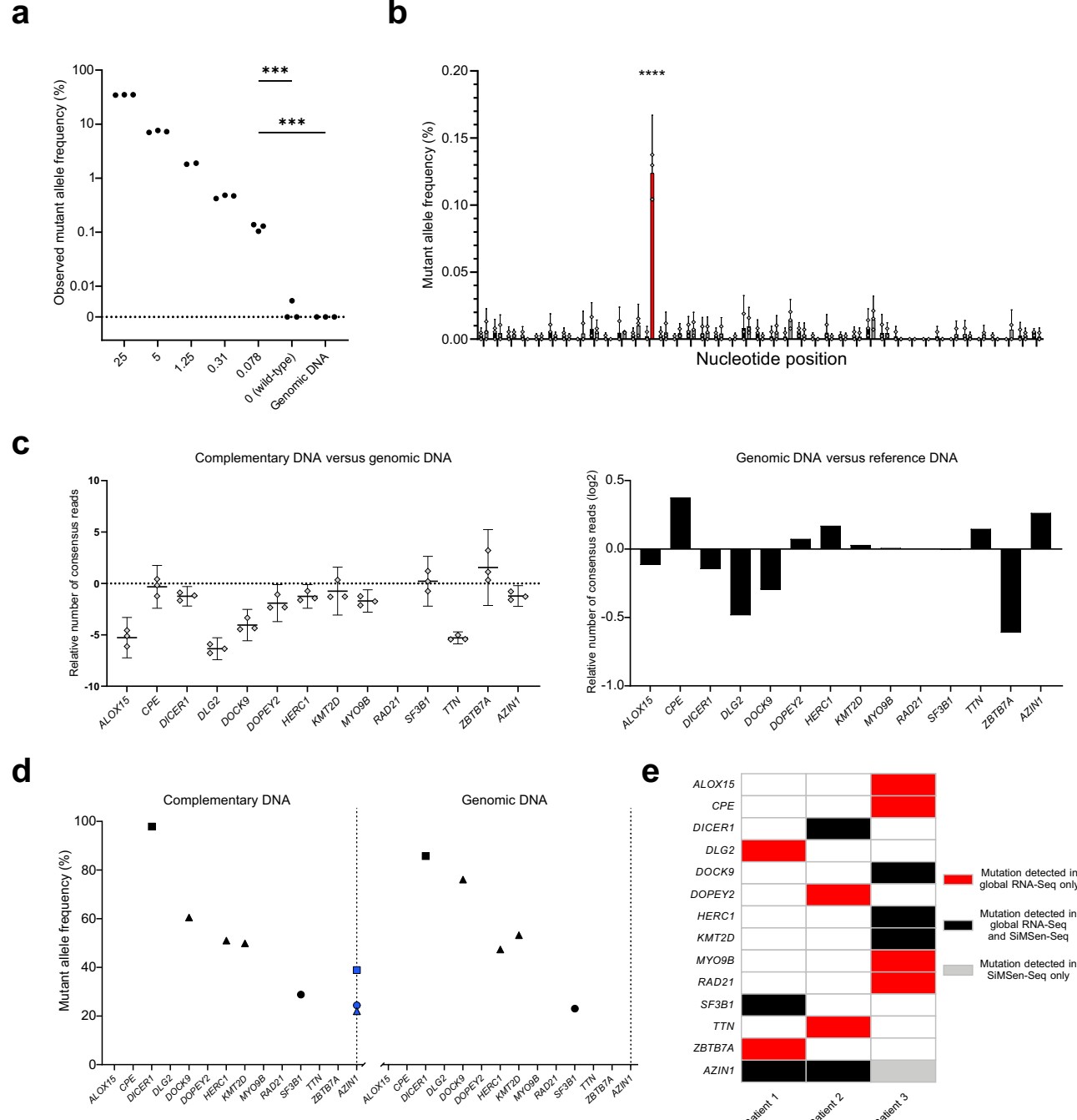

**Fig. 6 | Detection of mutations using digital RNA sequencing. a** Detection of spike-in *TP53* single nucleotide variant. The observed mutant allele frequency is shown at the y-axis, while the expected spike-in mutant allele frequency is shown at the *x*-axis. The *TP53* assay F, analyzing 200 ng total RNA or 10 ng genomic DNA was used, $n = 3$. ***$p = 0.0003$, unpaired Student's *t*-test. **b** Mutation detection and error rates. The error rates for samples with a spike-in mutation (gray) and controls without the spike-in mutation (black) samples are shown. The spike-in mutation position is indicated by red color with an expected mutant allele frequency of 0.078%. Mean + 95% CI is shown, $n = 3$. ****$p \leq 0.0001$, two-way ANOVA with Šidák correction. **c** Breast cancer panel cDNA yields. Relative number of consensus reads for breast cancer panel analyzing 200 ng total RNA, 20 ng matching genomic DNA as well as 20 ng reference DNA. Mean + 95% CI is shown for the left panel, $n = 3$. Mean only is shown for the right panel since no CI can be determined, $n = 3$. **d** Detection of breast cancer mutations. The observed mutant allele frequency is shown for cDNA and genomic DNA for three breast cancer samples. Note that the variation in *AZIN1* (blue), caused by RNA editing, is only detected at RNA level. **e** Summary of mutations called by global RNA sequencing (RNA-Seq) and SiMSen-Seq.

## Digital RNA sequencing enables ultrasensitive mutation detection and transcript profiling of cell-free RNA

To test the ability of our approach to detect low mutant allele frequencies we mixed total RNA from MLS 1765–92 cells with total RNA from MLS 402–91 cells that is heterozygous at one nucleotide position in *TP53* (Supplementary Fig. 7). The expected mutant allele frequencies ranged from 25–0.078%. Figure 6a–b show that all mutant allele frequencies were reliably detected and the number of detected molecules with mutations were significantly above the background of wild-type RNA as well as genomic DNA. Supplementary Fig. 8 shows that the samples with 0.078% mutant allele frequency only could be detected using UMIs.

To demonstrate the use of our approach, we designed a breast cancer panel, covering 15 mutations at 14 nucleotide positions previously identified with global RNA sequencing in three breast cancers[28]. Figure 6c shows the relative number of consensus reads for matching RNA and DNA samples. We detected higher number of molecules for two genes (*SF3B1* and *ZBTB7A*) at RNA level compared with DNA level, while ten times more molecules were detected at DNA level for four genes (*ALOX15*, *DLG2*, *DOCK9* and *TTN*). Supplementary Fig. 9a–c show the error rates and error correction factors for the breast cancer panel. We detected 7 out of 15 previously called mutations at RNA level (Fig. 6d, e). The *AZIN1* mutation was detected in patients 1 and 2, but unexpectedly also in patient 3. We detected the same mutations at DNA level, except that no *AZIN1* mutations were found. Interestingly, the mutation in *AZIN1* (S367G) is a known site for the ADAR protein family to RNA edit adenosine to inosine[29]. We concluded that 8 out of 15 mutations originally called as mutation using global RNA sequencing data were false positives, i.e., not a single molecule with the expected mutation was detected at either RNA or DNA level using digital RNA and digital DNA sequencing, respectively. Furthermore, to test the sensitivity to detect mutations at lower allele frequencies, we diluted the RNA sample from patient 1 with MLS 1765–92 RNA and detected mutated *SF3B1* transcripts even with a 1:64 dilution factor (Supplementary Fig. 9d).

We next extracted cell-free RNA from blood plasma collected from healthy individuals but the number of cell-free *TP53* transcripts were very low, resulting in non-reproducible sequencing libraries. Therefore, we developed a panel targeting five sequences in the abundantly expressed hemoglobulin subunit beta (*HBB*) gene (Supplementary Fig. 4). For comparison, we also analyzed genomic DNA. Figure 7a shows the number of detected *HBB* transcripts and Fig. 7b displays the number of *HBB* transcripts per milliliter blood plasma that varied between 2500 and 16000 molecules for the five *HBB* sequences. Next, we determined the mean error rates for all assays to be <0.005% (Fig. 7c), and the error correction factor using UMIs was 69.6 times (Fig. 7d). Similar values for error rates and error correction factor were observed for genomic DNA. Finally, we applied the hotspot mutation- and breast cancer panel to analyze cell-free RNA (Fig. 7e). We detected between 0 and 16 molecules per ml plasma, which is in the expected range for lowly and intermediately expressed genes in plasma[30]. As expected, no mutations were found in the liquid biopsy samples.

## Discussion

Digital RNA sequencing using UMIs offers improved possibilities to assess RNA molecules, including ultrasensitive mutation detection. In contrast to standard RNA sequencing protocols[31–33], our approach enhances mutation detection sensitivity through deep sequencing where all initial target sequences are sequenced multiple times and UMIs correct polymerase-induced errors. Deep sequencing can be applied to most approaches but is in practice restricted to targeted sequences. For example, a mammalian cell contains around 100,000 mRNAs and the exome consists of about 20 million base-pairs[34]. Assuming 5% of the exome is transcribed to the same degree, analyzing 1000 cells would require $10^{12}$ reads to cover all sequences once, given that each read covers a 100 nucleotides long sequence. To set this in context, the latest NovaSeq X Plus sequencer only generates up to about $10^{11}$ reads. Hence, deep sequencing is currently limited to applications that rely on targeted sequencing. In our study, we sequenced all targets molecules in the sample multiple times to maximize the sensitivity. The sensitivity is directly linked to the error rate since a mutation cannot be called with a lower allele frequency than the estimated error rate. The error correction factor is thus an estimate of the improvement provided by UMIs using deep sequencing data. We show that RNA molecules can be profiled with the same level of sensitivity to detect mutations as DNA analysis, i.e., <0.1%[21].

In our approach, we tag cDNA molecules with UMIs that will not compensate for errors occurring during RT. To compensate for all possible technical errors during library construction, the RNA molecules need to be tagged with UMIs before RT. However, adding UMIs to single-stranded RNA molecules, for example with ligation, is experimentally challenging. Some global RNA sequencing approaches incorporates UMIs in the oligo-dT primer[35]. These strategies also fail to compensate for errors introduced during the RT step and are not suitable for targeting sequences that are not directly upstream of the poly-A tail. Target-specific RT primers with UMIs may address this issue but multiplexing gene-specific RT primers containing UMIs is poorly studied, and this strategy still will not correct for errors occurring during RT. Our data also show that error rates for analyzing RNA molecules using optimal RT conditions are similar to those for genomic DNA molecules, suggesting that most errors occur after RT during library construction or sequencing. This is not surprising since target sequences are amplified numerous times during both library construction and cluster generation in the sequencer, while RNA molecules are only reverse transcribed once.

In SiMSen-Seq, target DNA or cDNA are tagged with UMIs during three barcoding PCR cycles. The number of errors that is potentially introduced during the PCR barcoding step is low[36] and other digital DNA sequencing approaches use up to 15 barcoding PCR cycles maintaining low error rates[37]. A disadvantage with increasing the number of barcoding PCR cycles is that the number of UMIs per target DNA molecule increases, which results in lower precision when estimating the original number of molecules in the sample[38]. Digital PCR[39], BEAMing[40] and in situ hybridization assays[41] are alternative approaches that offer detection of low mutant allele frequencies. However, these strategies are limited to the analysis of single or few target sequences where prior knowledge of RNA sequence variation is needed.

For ultrasensitive mutation detection in limiting sample types, such as fine needle aspirates and liquid biopsies, high cDNA yield is important since it increases the number of molecules that can be assessed and hence, the probability to find rare, mutated molecules. We observed large differences in cDNA yields between RT conditions as well as between assays targeting *TP53*. This variability was expected since secondary and tertiary RNA structures are known to affect cDNA yield[1,2]. Consequently, different RT conditions, including type of reverse transcriptase, reaction buffer, priming strategy and reaction temperature profile contribute to the cDNA yield. All tested RT conditions used a blend of random hexamers and oligo-dT as primers, which is considered optimal for high cDNA yield[42]. Ultrasensitive mutation analysis also requires low technical error rates, including both RT and library construction. Reverse transcriptase generates more errors than high fidelity DNA polymerases. The fidelity of the applied DNA polymerase is >300 times compared to standard *Taq* DNA polymerase that has reported fidelity between $1 \times 10^{-4}$ and $5 \times 10^{-5}$[43,44]. The reported fidelity of reverse transcriptases ranges between $1 \times 10^{-4}$ and $9 \times 10^{-5}$[45–47]. Substitution errors and sequence rearrangement, such as insertions and deletions, have been observed for both DNA polymerases and reverse transcriptases. The observed error rates include all types of errors that occur during nucleic acids extraction, RT, library construction errors, sequencing and bioinformatics as well as variants in RNA sequences that have a biological origin. We observed no direct link between cDNA yields and error rates. For example, Transcriptor generated the highest cDNA yield and also caused the most errors, while Superscript IV generated the lowest cDNA yield but caused the second most errors. Our UMI-based approach corrects for polymerase-induced errors but not for other types of technical errors that occur in the experimental steps before SiMSen-Seq, including RNA extraction and RT.

Here, we chose Omniscript for downstream analysis, since this RT condition generated high cDNA yield combined with low error rates. With this RT condition, the observed error rates varied between 0% and 0.04% among all targeted RNA sequences, which were at similar levels as the corresponding genomic DNA sequences, indicating that the error contribution of the RT step using Omniscript is at least at the same level as the errors generated during library construction and sequencing. A low error rate of RT is critical in our approach since the use of UMIs can only correct for errors that occur after cDNA synthesis. For example, Omniscript has about 7.5 times higher sensitivity than Superscript IV to identify mutations using the *TP53* assays. The dynamic range of Omniscript using different RNA concentrations was not linear throughout all tested RNA concentrations. This observation was expected and in line with other studies

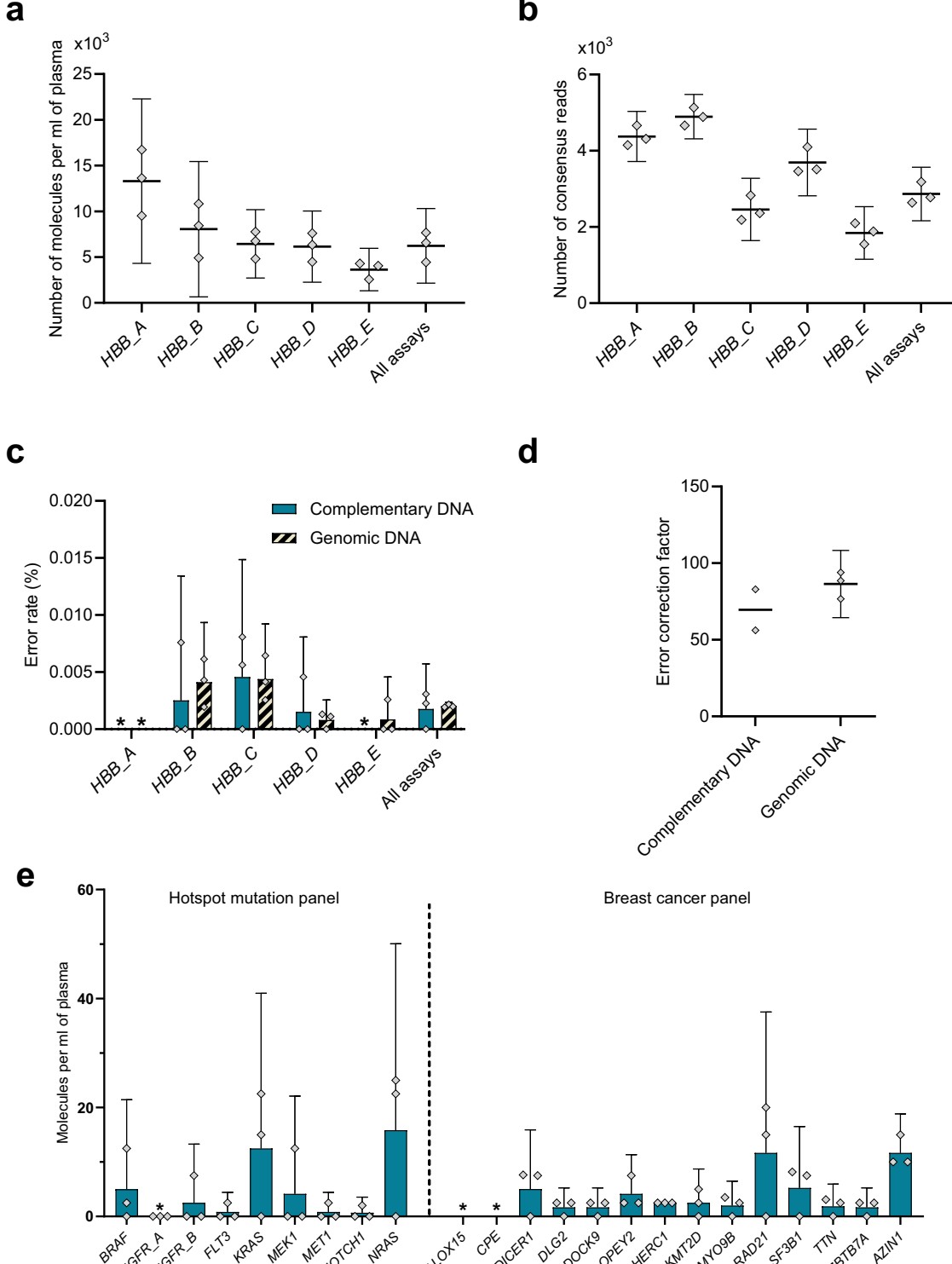

**Fig. 7 | Detection of cell-free RNA in blood plasma. a** Number of detected cDNA molecules per mL blood plasma calculated from consensus reads. The mean of all five assays is shown to the right. Mean + 95% CI is shown, $n = 3$. **b** Number of consensus reads for five *HBB* assays analyzing 10 ng genomic DNA as reference. The mean of all five assays is shown to the right. Mean + 95% CI is shown, $n = 3$. **c** Error rate per assay. The mean of all five assays is shown to the right. Mean + 95% CI is shown, $n = 3$. **d** Error correction factor using UMIs. The error rates before and after

using UMIs were calculated and compared. Mean + 95% CI is shown for genomic DNA, $n = 3$. For cDNA, $n = 2$ since no error was detected for one sample. Data for all nucleotide positions with and without UMIs are shown in Supplementary Fig. 10. * indicates no detected error. **e** Number of detected cDNA molecules per mL blood plasma for hotspot mutation- and breast cancer panel, respectively. Mean + 95% CI is shown, $n = 3$. * indicates no detected molecules.

evaluating RT conditions followed by quantitative PCR[2]. The RT linearity may potentially be improved by optimization of reaction conditions and the use of carriers. Interestingly, we observed differences in capacity to correct errors occurring after RT, i.e., variations in the error correction factor between the various RT conditions. We speculate that this variation may be due to properties of specific reverse transcriptases and their ability to reverse transcribe different RNA sequences as well as RNA modifications[45,46]. In conclusion, our data clearly show that the RT step is critical for ultrasensitive digital RNA sequencing and that the properties of different RT conditions are highly variable. More sequences need to be assessed to identify an overall optimal RT condition to provide maximal cDNA yields and minimal error rates.

In this study, we show that digital RNA sequencing using UMIs enables ultrasensitive mutation detection. Our approach is simple to perform and flexible in multiplexing. The performance is improved by choosing an optimal RT condition that provides both high cDNA yield in combination with low RT error rate, while the use of UMIs eliminates DNA polymerase-induced errors. We demonstrate that digital RNA sequencing has wide dynamic range and can be applied to detect low mutation allele frequencies as well as to analyze cell-free RNA molecules extracted from blood plasma. Ultrasensitive mutation analysis at RNA level is required in applications, such as assessment of transcriptional mutagenesis and RNA editing since these are not present at the DNA level. For mutations that exist at DNA level, it can be beneficial to use RNA, especially when the number of RNA molecules exceeds that of DNA molecules. To maximize the sensitivity to detect mutations, combined RNA and DNA analysis should also be considered. Digital RNA sequencing enables sensitive and specific RNA analysis in numerous basic research and clinical application areas, but additional studies are needed to determine clinical utility.

## Methods
### Cell cultures
The myxoid liposarcoma cell lines 1765–92 and 402–91 were cultured in complete media, containing RPMI 1640 GlutaMAX supplemented with 5% fetal bovine serum, 100 U/mL penicillin and 100 µg/mL streptomycin (all Gibco, Thermo Fisher Scientific). Cells were kept at 37 °C and 5% $CO_2$ and passaged using 0.25% trypsin with 0.5 mM EDTA (Gibco, Thermo Fisher Scientific). Sequencing showed that the cell lines contained no common dysfunctional or pathogenic mutations, including the in *TP53* gene[48]. All cell lines were routinely screened for mycoplasma infections using Mycoplasma PCR Detection Kit (Applied Biological Materials). Cell line authentication was performed with the cell line authentication test based on short tandem repeat analysis performed by Eurofins (Germany) and detection of cell line-specific fusion oncogenes at RNA level.

### Nucleic acids extraction
Total RNA was extracted using QIAcube with the RNeasy Micro Kit including DNase treatment (all Qiagen), according to the manufacturer's instructions. Genomic DNA was extracted using the AllPrep DNA/RNA/Protein Mini Kit (Qiagen), according to the manufacturer's instructions. RNA concentration was determined with a Qubit 3 Fluorometer using the Qubit RNA High Sensitivity Assay Kit (Invitrogen, Thermo Fisher Scientific). RNA integrity was assessed by capillary electrophoresis (Fragment Analyzer, Agilent Technologies) using the DNF-471 RNA Kit (Agilent Technologies), according to the manufacturer's instructions. The RNA quality numbers were >7.5 for all cell line samples. Total RNA was stored at −80 °C.

Circulating total RNA was isolated from blood plasma of healthy donors purchased from Zenbio. Plasma was mixed 1:1 with QIAzol Lysis Reagent (Qiagen) and incubated at room temperature for 5 min. The resulting solution was mixed with chloroform 9:2, followed by centrifugation at 12,000 *g* at 4 °C for 15 min. The clear upper phase was collected and RNA was extracted with Zymo RNA Clean & Concentrator kit with DNase treatment (Zymo Research), according to the manufacturer's instructions.

### Breast cancer tissue samples
Three primary invasive breast carcinomas that were previously analyzed for mutations at RNA level were used[28]. Genomic DNA and total RNA were extracted from the three fresh-frozen tissues where >70% neoplastic content in each sample was confirmed using touch preparation imprints stained with May-Grünwald Giemsa (Chemicon). Genomic DNA was isolated using the QIAamp Fast DNA Tissue Kit (Qiagen), while total RNA was isolated with the RNeasy Lipid Tissue Mini Kit (Qiagen), according to the manufacturer's instructions. The RNA concentration was assessed as described for cell line derived total RNA. DNA concentration was determined with Qubit 3 Fluorometer and High Sensitivity dsDNA Quantification Kit (Invitrogen, Thermo Fisher Scientific), while RNA integrity was assessed with Agilent 2100 Bioanalyzer using RNA 6000 Nano Kit (both Agilent Technologies), providing RIN values between 6.0 and 9.4.

### Reverse transcription
Reverse transcription was performed with seven different RT kits in a T100 Thermal Cycler (Bio-Rad Laboratories), according to the manufacturers' recommendations (Table 1). We used 400 ng total RNA for Transcriptor, Superscript IV, Primescript II and Goscript, 200 ng total RNA for Omniscript and Grandscript, and 50 ng total RNA for Sensiscript. We compensated for the variable amount of total RNA in either downstream SiMSen-Seq or in data analysis. Complementary DNA was diluted 1:1 with ultrapure distilled water (Invitrogen, Thermo Fisher Scientific) and stored at −20 °C. After initial evaluation of RT conditions, we used Omniscript for all downstream analysis. For the blood plasma analysis, the RNA concentration was below the limit of detection for Qubit. Hence, the used amount of cell-free RNA was <10 ng per RT reaction.

### Digital sequencing
Simple multiplexed PCR-based barcoding of DNA for ultrasensitive mutation detection using next-generation sequencing (SiMSen-Seq) was applied for digital sequencing[21]. All assays were short to enable detection of degraded nucleic acids. SiMSen-Seq library construction consists of two PCR steps followed by library purification. The first step comprised a 10 µL barcoding reaction with 0.1 U SuperFi Platinum Polymerase, 1 x SuperFi buffer, 200 µM dNTP mix (all Thermo Fisher Scientific), 0.5 M L-Carnitine (Sigma-Aldrich, Merck), 40 nM of each SiMSen-Seq barcoding primer (Ultramers, Integrated DNA Technologies, Supplementary Data 3) and either 2 µL diluted cDNA or 10–40 ng of human genomic DNA (Roche). The temperature profile was: 98 °C for 30 s; 3 cycles of amplification (98 °C for 10 s; 62 °C for 6 min; 72 °C for 30 s); 65 °C for 15 min; 95 °C for 15 min and hold at 4 °C. All ramping rates were 4 °C per sec. At the beginning of the 65 °C incubation step, 20 µL TE buffer (pH 8.0, Thermo Fisher Scientific) containing 30 µg/mL protease from *Streptomyces griseus* (Sigma-Aldrich, Merck) was added to terminate the reaction. The second 40 µL adapter PCR reaction contained 1 x Q5 Hot Start High-Fidelity Mastermix (New England BioLabs) and 400 nM of each Illumina adapter primer (desalted, Sigma-Aldrich, Merck) and 10 µL diluted barcoded PCR product. Standard Illumina adapter primer sequences were used[49]. The temperature profile was: 98 °C for 3 min; 30 cycles of amplification (98 °C for 10 s; 80 °C for 1 s; 72 °C for 30 s; 76 °C for 30 s, all with ramping at 0.2 °C/s) and hold at 4 °C.

Each library was analyzed by capillary electrophoresis using the Fragment Analyzer and the DNF-474 HS NGS DNA kit (Agilent Technologies), according to the manufacturer's instructions. Libraries were diluted based on their specific product concentrations using EB buffer (Qiagen) supplemented with 0.1% Tween (Sigma-Aldrich, Merck). Pooled libraries were purified with Pippin Prep using a 2% agarose gel cassette for 100–600 base pairs fragment sizes (both Sage Science). The purified library size was evaluated on a Fragment Analyzer using the DNF-474 HS NGS DNA kit. Sequencing was performed on either a MiniSeq using a High Output Reagent Kit or a NextSeq 550 using a NextSeq 500/550 Mid Output kit v2.5 (all Illumina). All sequencing were performed in single-end and 150 base pairs mode. Twenty percent PhiX Control v3 (Illumina) and a final library concentration of 1.8 pM was used in MiniSeq, while 10% PhiX and

1.3 pM library concentration was used in NextSeq 550. The sequencing depth per target sequence is shown in Supplementary Data 4.

Sequencing data was processed with UMIErrorCorrect version 0.29[50]. Unique molecular identifiers were trimmed off and then aligned to the reference genome. Next, reads were grouped into consensus reads based on UMI families to eliminate polymerase-induced errors. We applied a cut-off of at least three reads per UMI to generate consensus reads. The error rates were calculated as the total number of non-reference reads divided by the total number of reads per nucleotide position. Next, the mean for all positions in all assays was estimated. Nucleotide variants related to known single nucleotide polymorphisms, pseudogenes and RNA editing were excluded from analysis.

## Statistics and reproducibility
All experiments were performed in three replicates unless otherwise stated. Data are presented as mean ± 95% confidence interval (CI). Unpaired Student's *t*-test and two-way ANOVA with Šidák correction were used for statistical mutation analyses with GraphPad Prism 10.1.2 (GraphPad Software).

## Reporting summary
Further information on research design is available in the Nature Portfolio Reporting Summary linked to this article.

## Data availability
Raw sequencing data in FASTQ format is available at the NCBI Sequence Read Archive under submission ID PRJNA967219. The data behind the graphs in the manuscript and Supplementary Figures. are shown in Supplementary Data 5 and 6, respectively.

## Code availability
The UMIErrorCorrect pipeline with scripts is published[50] and available at Github: https://github.com/stahlberggroup/umierrorcorrect. No custom code is used.

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

## Acknowledgements

We thank Tony Godfrey for his useful comments and constructive criticism during the review of this manuscript. This research was funded by Assar Gabrielssons Research Foundation; Johan Jansson Foundation for Cancer Research; Region Västra Götaland, Sweden; Swedish Cancer Society (2022–2080); Swedish Childhood Cancer Foundation (2022–0030 and MTI2019–0008) Swedish Research Council (2021–01008); the Swedish state under the agreement between the Swedish government and the county councils, the ALF-agreement (965065); The Sjöberg Foundation and Sweden's Innovation Agency (2018–00421 and 2020-04141).

## Author contributions

M.L.S. and A.S. conceived and designed the experiments. T.Z.P. and K.H. provided clinical material. M.L.S. performed the experiments. M.L.S., D.A., T.Ö. and A.S. analyzed the data. M.L.S. and A.S. produced the figures. A.S. and M.L.S. wrote the paper. All authors critically reviewed the manuscript.

## Funding

## Ethical statement

All procedures related to breast cancer samples were performed in accordance with the Declaration of Helsinki and approved by the Medical Faculty Research Ethics Committee (# S164-02, Gothenburg, Sweden), where the ethical board waived the need of informed consent for retrospective biomarker analysis.

## Competing interests

A.S. is co-inventor of SiMSen-Seq that is patent protected (U.S. Serial No.:15/552,618). A.S. declares stock ownership and is board member in Iscaff Pharma, SiMSen Diagnostics and Tulebovaasta. All other authors declare no competing interests.
