## [Peer review file · Communications Biology]

Reviewers' comments:

Reviewer #1 (Remarks to the Author):

The authors developed a method for highly sensitive RNA sequencing by combining reverse transcription and UMIs. The authors compared yield and error rate, among other properties, on seven conditions in assays targeting six regions of TP53. For the best experimental condition, the authors showed that error rates were comparable to DNA sequencing. Lastly, the authors showed that the technique can be applied to plasma RNA to assess HBB. The comparison of the properties of the different RT conditions is useful and reasonably comprehensive but several properties of the optimal method, as well as the choice of genes assayed, leave some open questions regarding its potential applications.

1. The authors should provide the genomic coordinates of TP53 and HBB assayed. It is not clear where those regions are and how large those regions are. Many pathological specimens are stored FFPE, with potentially high levels of RNA degradation. The information on how large these assays are would be important to know if the technique is applicable for FFPE materials
2. There were no details on which sequencing machine was used, how deep the sequencing was, how long the reads were etc.
3. The authors did not evaluate error rates related to positions in sequencing reads and GC contents
4. RNA sequencing in the context of cancer (and cancer liquid biopsy) is limited as most cancer genes are tumor suppressor genes with reading-frame-alterations. Coupled with the fact that the RNA assay introduced insertions and deletions, it appears that its utility in the cancer setting is limited.
5. The authors chose TP53 as the exemplar gene for assay evaluation, which is fair enough given it is the most frequently mutated gene in cancer. Aside from the fact that many cancer-associated TP53 mutations may lead to nonsense-mediated decay (see point 4), TP53 gene itself is a fairly 'easy-to-sequence' gene, as in its capture is quite efficient and its alignment fairly unambiguous. The same cannot be said for many other cancer genes and mutations, such as RAS mutations and TERT promoter mutations. One could therefore argue that the assay was evaluated on the best case scenario.
6. Although the authors stated the potential utility of the RNA assay for plasma RNA, the fact that it only worked for HBB, a very, very abundantly expressed gene, is not convincing of its clinical potential.

Reviewer #2 (Remarks to the Author):

Detecting mutations using RNA sequencing is useful for different biological applications. The manuscript by Santamaría and Ståhlberg describes an optimized workflow for detecting rare nucleotide variants based on UMI-based targeted RNA sequencing. They test seven different reverse transcriptase (RT) enzymes and find Omniscript RT to be the best, based on cDNA yield and replicate variability. The cDNAs are then subjected to two barcoding steps for library preparation that add UMIs and sequencing adapters. These barcoding steps are previously used by the authors for detecting mutations based on the coding regions of genomic DNA (SiMSen-Seq; Ståhlberg et al., *Nucleic Acids Res.*, 2016). They use the UMI-tagged libraries to perform targeted sequencing of six different regions of cancer-relevant TP53 gene and find the error rates for certain RTs (Sensiscript, Omniscript, and Goscript) to be close to that observed for genomic DNA. The error rates are also

shown to be much lower for these RTs compared to the analysis done without UMI. The TP53 mutations are also reliably detected at their expected frequencies using this workflow. Finally, the authors demonstrate the application of this workflow by performing targeted-sequencing of HBB using cell-free RNA from blood plasma.

The workflow is promising, but is low-throughput. Global RNA-seq is generally good in detecting mutant alleles that are present at high frequency, but may not be good for detecting low frequency mutations. The authors demonstrate their workflow only for TP53, and not for other genes for which the mutation detection is not at all efficient using regular RNA-seq or other RNA-based methods for mutation detection. The broad applicability of this workflow is also a question as it is not clear if the assay conditions need to be optimized depending upon the gene. The cell-free RNA can be used for this workflow when the transcripts are expressed at high levels (TP53 did not work). The above concerns should be addressed along with the comments below.

Major comments:

1. The mutation detection efficiency for TP53 at RNA-level using this UMI-based workflow can be compared with the analysis done without UMIs. Is the UMI method better than the non-UMI one?
2. The mutation detection frequency with and without UMI can be compared for cell-free RNA (HBB gene) to see if the UMI method is better than the non-UMI one.
3. The authors should show how their method for detecting clinically-relevant mutations using RNA is better than the global RNA-seq or other RNA-based methods. They can use genes such as TERT for which mutations are not reliably detected by RNA-seq (see PMID: 32948110; also PMID: 30083469).

Minor comments:

1. The authors should compare their method with other published RNA-based methods for mutation detection out there. They should clearly explain where this method excels and do not excel compared to the published methods.
2. A clear workflow describing how the consensus reads are used to eliminate polymerase-induced errors and how error rate is determined will be beneficial for the readers.
3. The authors should show all the code used for different error-corrections and mutation detection. They can deposit the code on Github and include a link for that.

Reviewer #3 (Remarks to the Author):

The manuscript entitled “Digital RNA sequencing using unique molecular identifiers” by Santamaría and Ståhlbert reports on the application of a previously developed method SiMSen-Seq for UMI-enabled PCR amplicon resequencing on DNA, now applied to RNA. The manuscript is well written and informative, and the conclusions are supported by the results and figures.

major comments:

1. While this manuscript demonstrates that the SiMSen-Seq method also works on single-stranded cDNA, the manuscript does not seem to bring major new insights nor advances. On the one hand, it

is well known that different reverse transcriptases have different efficiencies and fidelity; on the other hand, error-correction using the SiMSen-Seq method is well document before.

2. Did the authors consider to make double-stranded cDNA? Would this bring advantages to error-correction?

3. While the TP53 gene is an import cancer gene, this reviewer questions the relevance of targeted mutation analysis of the HBB transcript. Also, why are these specific regions/amplicons selected in the TP53 and HBB transcript?

4. More generally, could the authors elaborate for which genes/transcripts they consider mutation analysis to be as / more relevant as DNA (the gold standard)? In line with this, in the introduction, the authors mention 'emerging RNA sequencing applications require the ability to detect low variant allele frequencies'. Which applications? Why?

5. Line 218: What is the relevance of the error-correction factor? What if an RT enzyme only generates very few errors? Then nothing to correct, but it remains a good enzyme, no? What contributes to this factor? Sequencing depth, RT error rate, yield?

6. Could the authors speculate on the reasons of the (varying) RT error rates? At least it seems that yield is not contributing. Could the authors compare their observed error-rates with what is known for RT enzymes in the literature?

7. 200 ng total RNA input is a very large amount. In clinical practice, such yields are difficult to obtain (especially for liquid biopsies). The authors should discuss the impact of RNA input amounts on limit of detection of variant allele frequencies and maximal error correction levels?

minor comments:

1. The use of the word 'digital' could be questioned and may be more fancy than informative. In fact, the method relies on plain massively parallel sequencing, around for more than a decade.

2. In the abstract: Error-free may be an overstatement; 'low error' is more appropriate.

3. p3: UMI do not eliminate quantification bias, but reduce it. Apart from amplification bias, also other quantification biases exist that the authors may highlight.

4. p3: Please rephrase: "Sequencing also generates polymerase-induced errors." Sequencing data contains such errors, but not generate it?

5. A space is needed between the unit and the value (e.g. 80 °C)

6. Line 110: total 'RNA' (add RNA)

7. I prefer to see 95% confidence intervals instead of standard deviations. CI are more informative to the reader.

8. What is the relevance of Figure 4C?

9. Why is assay 1 empty in Figure 7C?

Reviewer #1 (Remarks to the Author):

The authors developed a method for highly sensitive RNA sequencing by combining reverse transcription and UMIs. The authors compared yield and error rate, among other properties, on seven conditions in assays targeting six regions of TP53. For the best experimental condition, the authors showed that error rates were comparable to DNA sequencing. Lastly, the authors showed that the technique can be applied to plasma RNA to assess HBB. The comparison of the properties of the different RT conditions is useful and reasonably comprehensive but several properties of the optimal method, as well as the choice of genes assayed, leave some open questions regarding its potential applications.

We thank the reviewer for the positive and constructive feedback that helps us to improve the manuscript. We have included new data demonstrating that we can analyze multiple genes targeting positions for hotspot mutation and that we can do so even using very low amounts of total RNA. We have also clarified several experimental details and discuss when our approach is beneficial in relation to other strategies as well as its limitations. All new and updated Figures are added at the back of the response letter.

1. The authors should provide the genomic coordinates of TP53 and HBB assayed. It is not clear where those regions are and how large those regions are. Many pathological specimens are stored FFPE, with potentially high levels of RNA degradation. The information on how large these assays are would be important to know if the technique is applicable for FFPE materials.

Agreed, the genomic positions of assays are shown in Supplemental Table 1, including total amplicon length. Yes, the assay length is important. All designed assays are relatively short (between 63 and 102 bp) to enable analysis of fragmented RNA, such as present in FFPE materials. This information is added in Materials and methods (Lines 139-140) as well as in Supplemental Table 1.

2. There were no details on which sequencing machine was used, how deep the sequencing was, how long the reads were etc.

We have updated the Materials and methods (Lines 163-169) with detailed information about sequencing (instrument, kit, read length, etc.). All sequencing was performed as deep sequencing where all target molecules in the sample was analyzed multiple time. We have also included information about raw sequencing depth for all sample in Supplemental Table 2.

3. The authors did not evaluate error rates related to positions in sequencing reads and GC contents

Good point. We have now included a new figure showing the observed errors in relation to nucleotide position (Supplemental Figure 5). We could not identify any correlations between error rate and sequence context, including GC-content, in our data. After UMI correction, we have very few errors, where most nucleotide positions have no errors. We have updated the Discussion about expected error rates in our data (Lines 381-389). In future studies with more variable sequences, such as divergent GC

contents, it will be interesting to study error rates in relation to sequence context. Note that we discovered a mistake in our estimate of raw read errors where we previously have underestimated the errors not using UMIs, resulting in updated Figures 2, 4D, 5B and 7D as well as Supplemental Figure 4. We have pointed out that more analyses are needed to make stronger conclusions (Lines 411-413).

4. RNA sequencing in the context of cancer (and cancer liquid biopsy) is limited as most cancer genes are tumor suppressor genes with reading-frame-alterations. Coupled with the fact that the RNA assay introduced insertions and deletions, it appears that its utility in the cancer setting is limited.

Indeed, this is an important point. Our data indicate that there may be an increase of false InDels for worse performing RT conditions. However, the overall error rate for the well-performing RT conditions does not generate more errors compared with DNA analysis. Furthermore, we do not detect more InDels than substitutions for any data. InDels are also well-known problems for DNA analysis, especially repetitive sequences. For example, stutter formation in forensics STR analysis is a common application that is biased by InDels. It should be noted that the error rates for all our RT conditions using UMIs is far below the commonly applied cut-offs (often between 1% and 5%) for mutation analysis in both DNA and RNA without UMIs. It is at the single molecule level we observe more InDels for specific RT conditions compared to DNA analysis. We have extended our Discussion (Lines 380-389, 393-400 and 414-428) with information about error rates and the use of our approach.

Furthermore, for applications, such as transcriptional mutagenesis and post transcriptional RNA editing, analysis can only be performed at RNA level since no variations exist at the DNA level. We have now included more data in our results (hotspot mutation panel and breast cancer panel) to demonstrate the benefits and feasibility of using our strategy (Lines 236-242, 261-269 and 294-311 and 321-325). In addition to mutations that exist at both DNA and RNA level, we also include analysis of RNA editing in the AZIN1 gene (Lines 302-305). We have also demonstrated that the new panels can be applied in liquid biopsies. Finally, we have updated our discussion to highlight when mutation analysis at RNA level is potentially useful in comparison with DNA (last paragraph).

5. The authors chose TP53 as the exemplar gene for assay evaluation, which is fair enough given it is the most frequently mutated gene in cancer. Aside from the fact that many cancer-associated TP53 mutations may lead to nonsense-mediated decay (see point 4), TP53 gene itself is a fairly 'easy-to-sequence' gene, as in its capture is quite efficient and its alignment fairly unambiguous. The same cannot be said for many other cancer genes and mutations, such as RAS mutations and TERT promoter mutations. One could therefore argue that the assay was evaluated on the best case scenario.

This is relevant criticism. Hence, we have developed more panels including more assays. First, we developed a hotspot mutation panel, targeting commonly mutated sites in BRAF, EGFR, FLT3, KRAS, MEK1, NOTCH1 and NRAS. This panel performed well and we included this data in Figures 3E, 4E and 4F as well as all assay information in Supplemental Table 1. Next, we also developed a breast cancer panel consisting of 14 assays, to assess previously identified mutations in three breast cancer samples. This panel also performed well and data are shown in Figures 6C-E and Supplemental Figures 7B-E. In addition to mutation identification at both DNA and RNA level, we also detected one variant that is caused by RNA editing (AZIN1, a known RNA-edited gene, edited by the ADAR protein family) that can only be detected at RNA level. Non-transcribed DNA, like TERT promoters cannot be assessed at RNA level. We have also updated Materials and methods (mainly lines 112-124) and Results (Lines 236-242, 261-269 and 294-311 and 321-325) with all new data.

6. Although the authors stated the potential utility of the RNA assay for plasma RNA, the fact that it only worked for HBB, a very, very abundantly expressed gene, is not convincing of its clinical potential.

This is true. We choose HBB due to the fact it is highly expressed to demonstrate the error-correction capacity of the method, an analysis that require large number of molecules. To demonstrate feasibility, we have now analyzed cell-free RNA with both the hotspot mutation panel and the breast cancer panel using control plasma from healthy individuals. The panels performed good and most genes were expressed to a low level (in line with published data PMID: 33883548). These data are now shown in Figure 7E, which show that we can apply our approach to more challenging sample types with limited number of molecules (Lines 321-325). We have also demonstrated that the hotspot mutation panel can be successfully applied down to 1 ng total cell line RNA (Figure 3E and lines 236-242).

Reviewer #2 (Remarks to the Author):

Detecting mutations using RNA sequencing is useful for different biological applications. The manuscript by Santamaría and Ståhlberg describes an optimized workflow for detecting rare nucleotide variants based on UMI-based targeted RNA sequencing. They test seven different reverse transcriptase (RT) enzymes and find Omniscript RT to be the best, based on cDNA yield and replicate variability. The cDNAs are then subjected to two barcoding steps for library preparation that add UMIs and sequencing adapters. These barcoding steps are previously used by the authors for detecting mutations based on the coding regions of genomic DNA (SiMSen-Seq; Ståhlberg et al., Nucleic Acids Res., 2016). They use the UMI-tagged libraries to perform targeted sequencing of six different regions of cancer-relevant TP53 gene and find the error rates for certain RTs (Sensiscript, Omniscript, and Goscript) to be close to that observed for genomic DNA. The error rates are also shown to be much lower for these RTs compared to the analysis done without UMI. The TP53 mutations are also reliably detected at their expected frequencies using this workflow. Finally, the authors demonstrate the application of this workflow by performing targeted-sequencing of HBB using cell-free RNA from blood plasma.

The workflow is promising, but is low-throughput. Global RNA-seq is generally good in detecting mutant alleles that are present at high frequency, but may not be good for detecting low frequency mutations. The authors demonstrate their workflow only for TP53, and not for other genes for which the mutation detection is not at all efficient using regular RNA-seq or other RNA-based methods for mutation detection. The broad applicability of this workflow is also a question as it is not clear if the assay conditions need to be optimized depending upon the gene. The cell-free RNA can be used for this workflow when the transcripts are expressed at high levels (TP53 did not work). The above concerns should be addressed along with the comments below.

We thank the reviewer for the positive and constructive feedback, contributing to an improved manuscript. We have now included new data, using a hotspot mutation panel and a breast cancer panel, targeting in total 21 additional genes that are commonly mutated in cancer. We also successfully demonstrated that the same panels can be used for cell-free RNA analysis. We have clarified the use of our approach to detect variants at RNA level in relation to DNA level and we also discuss when our approach is beneficial to use. All new and updated Figures are added at the back of the response letter.

Major comments:

1. The mutation detection efficiency for TP53 at RNA-level using this UMI-based workflow can be compared with the analysis done without UMIs. Is the UMI method better than the non-UMI one?

Indeed, this is a relevant point to visualize. Supplemental Figure 7 (Lines 292-293) illustrates that the lowest mutant allele frequency cannot be detected without the use of UMIs. Overall, the error rate and the error correction factor are directly correlated to the lowest mutant allele frequency that can be detected and how much the use of UMIs improve the data, respectively. Our approach utilizes two concepts to achieve high sensitivity and specificity, (i) deep sequencing where we sequence all molecules in each sample multiple times, and (ii) UMIs that correct polymerase-induced errors. We have clarified the definition of error rates and error correction factor in the Results section (Lines 207-211 and 257-259) and updated the discussion clarifying the concepts (Lines 329-345). For more details about mutant calling using UMI-based sequencing data, such as SiMSen-Seq data, see PMID: 36031761. Note that we discovered a mistake in our estimate of raw read errors where we previously have underestimated the errors not using UMIs, resulting in updated Figures 2, 4D, 5B and 7D as well as Supplemental Figure 4.

2. The mutation detection frequency with and without UMI can be compared for cell-free RNA (HBB gene) to see if the UMI method is better than the non-UMI one.

Agreed, we have added HBB data with and without UMIs in Supplemental Figure 8. See also response to comment 1.

3. The authors should show how their method for detecting clinically-relevant mutations using RNA is better than the global RNA-seq or other RNA-based methods. They can use genes such as TERT for which mutations are not reliably detected by RNA-seq (see PMID: 32948110; also PMID: 30083469).

This is an insightful comment. In our setup we improve sensitivity and specificity compared with standard RNA sequencing using deep sequencing in combination with UMIs. We have clarified this in the discussion (Lines 67- 73 and 329-345). In standard RNA sequencing you could theoretically apply deep sequencing but practically it is not feasible. Normally, global RNA sequencing is used to identify mutations at low coverage. Then, selected mutations are verified with deep sequencing, not using UMIs (PMID: 31171663 is one example of this strategy). We show that we can detect mutations with mutant allele frequency < 0.1% at RNA level (in most cases also << 0.1%). This is the same range as for DNA analysis using UMIs (PMID: 31391323, 27060140 and 22853953) (Lines 355-360).

To enable detection of mutant allele frequency < 0.1%, at least 1000 molecules for a specific target sequence are required in average. In standard RNA sequencing, this sensitivity is rarely achieved since most target sequences only provide data for a few molecules (<< 1000 molecules). For example, a mammalian cell contains about 100,000 mRNAs and the exome consists of about 20 million base-pairs. If we analyze 1000 cells and assume that only 5% of the exome is transcribed to the same degree. Then, if each read covers a 100 nucleotides long sequence, we would need 10^{12} reads to assess all sequences. To set this in context, the latest NovaSeq X Plus sequencer from Illumina generates up to about 10^{10} reads. Hence, deep sequencing is currently only practically useful for applications requiring targeted sequencing, while normal global RNA sequencing cannot be used for ultrasensitive mutation analysis. We have updated the discussion with this information (including references, Lines 329-345), clarifying the limitations with global RNA sequencing and the advantageous with our setup.

We also performed global RNA sequencing using the SMART-Seq2 protocol (PMID:24385147) with minor modifications analyzing total RNA extracted from MLS 1765-92 cells. We sequenced four samples with an average sequencing depth of 26 million reads per sample. Here, we detected 16.2 million nucleotide position with at least 10 reads. However, only 5 % and 0.3% of all detected nucleotide positions had > 100 reads and > 1000 reads, respectively. This again, illustrates that global RNA sequencing is not suitable for deep sequencing. Furthermore, it shows that only a very small proportion of all transcripts are detected numerous times. These new data are not included in this manuscript since it is in line with other published studies by us and others. However, we are open to include these data if the reviewer believes it add value to our study.

Digital PCR, BEAMing and in situ hybridization assays can also be used to detect rare mutations at RNA levels. These strategies are limited to single or at best for a few target sequences, where the exact mutation needs to be known in advance. We have updated the discussion with information about these methods (and references) and its relation to our approach (Discussion, Lines 367-370)

The clinical use of RNA in somatic mutation analysis is restricted to the exome. Hence, areas such as TERT promoters and other non-transcribed DNA regions cannot be assessed. In contrast, transcriptional mutagenesis and post transcriptional RNA editing can only be assessed at RNA level. Hence, there are several applications that are analyte dependent. There are also some applications that benefits from combined analysis. We have updated the introduction, highlighting the use of DNA and RNA analysis, respectively (Lines 45-57).

Minor comments:

1. The authors should compare their method with other published RNA-based methods for mutation detection out there. They should clearly explain where this method excels and do not excel compared to the published methods.

We have updated the discussion about our approach in relation to standard RNA sequencing methods and how deep sequencing and UMIs contribute to the improved sensitivities with our strategies. We have also included information about alternative methods to assess RNA, including digital PCR, BEAMing and in situ hybridization assays, that may provide similar sensitivities to detect mutations. See also response to the last comment 3 above (Lines 328-345 and 361-370).

2. A clear workflow describing how the consensus reads are used to eliminate polymerase-induced errors and how error rate is determined will be beneficial for the readers.

Agreed, we have now included two new figures (Supplemental Figures 1A - B) that illustrate the concept of UMIs and bioinformatics pipeline, respectively. We have also clarified how the error rate was calculated in the Result section (Lines 205-211). We have also included a new Supplemental Table 3 with the count matrices for data with and without UMI correction using a part of the data used in Figure 2 as example from which the error rate and error correction factor can be calculated.

3. The authors should show all the code used for different error-corrections and mutation detection. They can deposit the code on Github and include a link for that.

Agreed, the bioinformatics pipeline, UMIErrorCorrect, is open-source code. The code is available at GitHub: <https://github.com/stahlberggroup/umierrorcorrect>. Here, we also provide a tutorial for the pipeline. We have included a Code Availability Statement with this information (Lines 438-439).

Reviewer #3 (Remarks to the Author):

The manuscript entitled “Digital RNA sequencing using unique molecular identifiers” by Santamaría and Ståhlbert reports on the application of a previously developed method SiMSen-Seq for UMI-enabled PCR amplicon resequencing on DNA, now applied to RNA. The manuscript is well written and informative, and the conclusions are supported by the results and figures.

We thank the reviewer for the positive and informative comments, resulting in an improved manuscript. We have updated the manuscript with new data demonstrating the feasibility and use of our approach. In addition, we have also clarified data and concepts related to the provided comments. All new and updated Figures are added at the back of the response letter.

major comments:

1. While this manuscript demonstrates that the SiMSen-Seq method also works on single-stranded cDNA, the manuscript does not seem to bring major new insights nor advances. On the one hand, it is well known that different reverse transcriptases have different efficiencies and fidelity; on the other hand, error-correction using the SiMSen-Seq method is well document before.

Indeed, it is documented that SiMSen-Seq enables rare mutant detection at DNA levels with high error correction. Separately, it is also known that reverse transcriptases have variable properties, providing different cDNA yield and errors. However, there is limited knowledge about applying an ultrasensitive sequencing protocol that enable detection of multiple target sequences, such as SiMSen-Seq, at RNA level. We are not aware of any study using a strategy that overlaps with our approach. Furthermore, our findings about the RT conditions in relation to mutation detection should be informative to a wide research community since it can easily be adapted to any targets of interest. Our setup provides previously unknown information about cDNA yield and error rates that are important to consider, including the importance of careful protocol optimization. We have updated the study with new data, demonstrating the benefits our approach offers (Lines 236-242, 261-269, 294-311, 321-325 and Figures 3E, 4E-F, 6C-E and 7E as well as Supplemental Figures 1, 5-8).

2. Did the authors consider to make double-stranded cDNA? Would this bring advantages to error-correction?

Indeed, this is an interesting possibility and will most likely work. However, with the SiMSen-Seq approach this would not bring any advantages since double-stranded cDNA will have one molecule extra generated before incorporation of UMIs, which is an additional source of technical errors that cannot be corrected in the current setup. An alternative strategy is to add one additional cycle in the barcoding PCR of SiMSen-Seq. However, none of our data indicate that this will improve the protocol. We discuss the effects of variable barcoding cycles in the Discussion (Lines 361-370).

3. While the TP53 gene is an import cancer gene, this reviewer questions the relevance of targeted mutation analysis of the HBB transcript. Also, why are these specific regions/amplicons selected in the TP53 and HBB transcript?

Target sequences in TP53 are covering commonly mutated nucleotide positions. In HBB, we selected sequence to be in different exons since the gene has no specific hotspot mutation positions. We have updated the legend with this information (We moved the old Figure 1B to become Supplemental Figure

3). We have now also included data for a hotspot cancer mutation panel, targeting seven genes that are commonly mutated as well as a breast cancer panel targeting previously identified mutations that we used to analyze three breast cancer samples at both RNA and DNA level. Information and data about these new panels and assays are described in Results (Lines 236-242, 261-269, 294-311, 321-325), Figure 3E, 4E-F and Supplemental Table 1.

4. More generally, could the authors elaborate for which genes/transcripts they consider mutation analysis to be as / more relevant as DNA (the gold standard)? In line with this, in the introduction, the authors mention 'emerging RNA sequencing applications require the ability to detect low variant allele frequencies'. Which applications? Why?

This is an insightful comment. We have included new experimental data, demonstrating how mutations can be assessed at both DNA and RNA level (Figures 6C-E and Lines 294-311). These data also included data for post transcriptional RNA editing, which can only be detected at RNA level. We have updated our discussion, outlining when DNA and/or RNA analysis is preferred (Lines 420-426). For example, when the number of RNA molecules is higher than the number of DNA molecules it may be beneficial to perform mutation analysis at RNA level. Furthermore, combined DNA and RNA analysis may also be beneficial to use.

We have also updated and clarified the introduction about the use of mutation analysis at RNA level, where application related to transcriptional mutagenesis and post transcriptional RNA editing can only be analyzed at RNA level (Lines 47-57).

5. Line 218: What is the relevance of the error-correction factor? What if an RT enzyme only generates very few errors? Then nothing to correct, but it remains a good enzyme, no? What contributes to this factor? Sequencing depth, RT error rate, yield?

This is an important point. We have clarified the definition of error rate and error correction factor in the manuscript (Results, Lines 205-211 and 257-259). The absolute error rate level is the total overall error that remains after using UMIs, including errors introduced during pre-analytics, RT step, library construction, sequencing and bioinformatics, as well as true biological variations. The error correction factor is how many times in deep sequencing data we can reduce the error only by the use of UMIs. In all our analysis, essentially all errors are corrected, i.e., high error correction factor. We cannot correct for errors that occur during the RT step. Therefore, reverse transcriptases with high fidelity are generally preferred when they provide acceptable cDNA yield. In all our analyses we cannot separate out a specific RT error for well-performing RT conditions, since the error rates for DNA and RNA analysis are essential at the same level. However, for some RT conditions, the error rate is higher. Still, it will be significantly better to use our setup for mutation analysis at the RNA level compared with normal RNA sequencing, including deep sequencing, since they do not offer any error correction. We have updated the Discussion with detailed information about error rates for RT, such as reported fidelities for reverse transcriptases as well as DNA polymerases (Lines 380-392 and 394-402). We also discuss why we observe differences between RT conditions in relation to enzymatic properties and the ability of reverse transcriptase to transcribe different sequence contexts as well as RNA modifications (Lines 405-409). Our data show that RT yield is not related to error rate (Figure 4C). The sequencing depth is directly proportional to the number of molecules per target since we analyze all target sequences that exist in the sample multiple times. In Supplemental Table 2, we show raw sequencing depths for all samples and in Supplemental Table 3 we exemplify some data at specific nucleotide positions, with and without error correction.

6. Could the authors speculate on the reasons of the (varying) RT error rates? At least it seems that yield is not contributing. Could the authors compare their observed error-rates with what is known for RT enzymes in the literature?

The error rate for reverse transcriptase is about 100 times higher compared with high fidelity DNA polymerases, according to literature, where substitutions are more common than other DNA rearrangement. One possible explanation for differences between RT conditions is the specific enzymatic properties of reverse transcriptases and how they transcribe different sequences, including RNA modification. There are reports showing variations in error rates among reverse transcriptases and that some have lower tendency to generate InDels than others (see review PMID: 32653101) and there are also some literatures around reverse transcriptase fidelity but none has to our knowledge defined any underlying mechanism in detail. It is challenging to compare error rates between studies, since essentially all methods assess the error rates indirectly, including our setup. In our data, most errors for the well-performing RT conditions that remain after UMI correction are still due to the DNA polymerase and not due to the RT step. However, our observed error rates are within the expected range for reported fidelities (Figures 4, 5 and 7 as well as Supplemental Figure 7), although with some variations among specific assays and target sequences. We have updated the Discussion about reverse transcriptase and DNA polymerase in relation to their fidelities with numbers as well as information regarding reverse transcriptases having different properties (Lines 380-392, 405-409). We have also clarified the sources to errors in our analyses (Lines 208-211 and 387-389). See also response to comment 5 above. Note that we discovered a mistake in our estimate of raw read errors where we previously have underestimated the errors not using UMIs, resulting in updated Figures 2, 4D, 5B and 7D as well as Supplemental Figure 4.

7. 200 ng total RNA input is a very large amount. In clinical practice, such yields are difficult to obtain (especially for liquid biopsies). The authors should discuss the impact of RNA input amounts on limit of detection of variant allele frequencies and maximal error correction levels?

Indeed, this is an important aspect. We used large amount of total RNA in several experiments to assess the ability to detect variant alleles at low frequencies. In figure 3D we show that lower amounts of total RNA can be used reproducibly. We have also added new experimental data using a new hotspot mutation panel (Lines 236-242). In Figure 3E, we demonstrated that we can easily assess 200, 5 and 1 ng total RNA with this hotspot mutation panel. We also performed additional liquid biopsy analysis using two different gene panels (hotspot mutation panel and breast cancer panel). Here, we used < 10 ng total RNA and analyzed low expressed genes (Lines 320-325). Collectively, these new data show that low amount of total RNA can be used with our approach.

Most likely we have an even higher error correction factor than reported here since we have no observed errors for most nucleotide positions. To assess if this correction factor may be even higher, larger amounts of total RNA need to be assessed. However, the clinical relevance of this is not clear and beyond the scope of this article. The limit of detection for mutant detection is target dependent, since the limiting factor is related to the expression of each gene in most scenarios (For DNA analysis, it is simpler to assess limit of detection, since most DNA sequences are single locus). This is the reason we used high RNA concentrations in our initial experiments. In all our data, we never observed any error rate above 0.1%. and for the well-performing RT conditions, we did not observe higher error rates for RNA analysis compared with DNA analysis. We now discuss the effects of error rates in the Discussion at multiple places (for example: 380-413).

Note that if we have insufficient number of RNA molecules, the theoretical limit of mutant detection decreases as it would for corresponding DNA analyses. For example, for low-expressed RNA molecules (as well as for DNA molecules) in liquid biopsies the theoretical limit of detection is 5% if we in total only can detect 20 molecules.

minor comments:

1. The use of the word 'digital' could be questioned and may be more fancy than informative. In fact, the method relies on, around for more than a decade.

We agree to some degree, the word digital refer to the ability to analyze molecules one by one. For PCR methods we use digital PCR to count individual molecules. In some UMI-based sequencing-based approaches, such as the one used here, we also look at one molecule at the time subsequently to collapsing the raw sequencing reads. Hence, we believe that referring to our approach as 'digital sequencing' still has its merits. However, we are open to change the wording if needed.

2. In the abstract: Error-free may be an overstatement; 'low error' is more appropriate.

Agreed, we have changed to 'low error' in the abstract and end of introduction (Lines 23 and 87).

3. p3: UMI do not eliminate quantification bias, but reduce it. Apart from amplification bias, also other quantification biases exist that the authors may highlight.

Agreed, we have changed from 'eliminating' to 'reducing'. Indeed, there are other sources of amplification biases, but we are not familiar of any additional quantification biases that are corrected by the use of UMIs. Otherwise, preanalytical factors such as extraction efficiency, RT efficiency and the ability of SiMSen-Seq primers to detect original molecules are examples of factors that confound RNA quantification in general, which are confounding factors shared with most sequencing protocols. We have pointed out that UMIs only correct for errors occurring after UMI labeling (Lines 346-360).

4. p3: Please rephrase: "Sequencing also generates polymerase-induced errors." Sequencing data contains such errors, but not generate it?

Agreed, polymerase-induced errors occur during library construction as well as during the cluster generation. We have re-phrased the sentence (Lines 62-64).

5. A space is needed between the unit and the value (e.g. 80 °C)

Updated (multiple places)

6. Line 110: total 'RNA' (add RNA)

Updated (Line 129)

7. I prefer to see 95% confidence intervals instead of standard deviations. CI are more informative to the reader.

We have updated the manuscript with 95% confidence interval for all relevant figures (Figures 2-7 as well as Supplemental Figures 4, 7 and 8).

8. What is the relevance of Figure 4C?

This figure shows that the number of detected errors increase with the number of analyzed molecules. In principle it correlates data in Figure 3A with 4A. This demonstrates that the error rate is not correlated to cDNA yield (discussed in major comment 5). We believe this is relevant to show but we are open to move the Figure to the supplement if recommended.

9. Why is assay 1 empty in Figure 7C?

Assay 1 showed zero errors after UMI correction. We have clarified this in the legend (we have also included this info in all new figures with 'zeros').

Figures

Figure 1. Schematic overview of SiMSen-Seq and assays. A) The SiMSen-Seq workflow for RNA and DNA analysis, respectively. Complementary DNA is single-stranded, while genomic DNA is double-stranded, resulting in three and six different UMIs, respectively. One-third of the reaction volume of barcoding PCR is loaded into the adapter PCR. The tested RT conditions are shown.

Figure 2. Digital sequencing using UMIs. Examples of sequencing data with and without error correction using UMIs. Total RNA and genomic DNA were analyzed from the same MLS 1765-92 cell line. Each position represents a specific nucleotide position in assay B, *TP53*. The error rate per position was calculated as the total number of non-reference reads divided by the total number of detected reads with and without UMI-error correction. Mean + 95% CI is shown, n = 3. The count matrices for these data with and without UMI-error correction is shown in Supplemental Table 3.

Figure 3. Reverse transcription properties. **A)** Complementary DNA yield. For RNA analysis one consensus read corresponds to one cDNA molecule, while two consensus reads correspond to one genomic DNA molecule. The Sensiscript cDNA yields are scaled by a factor of four to compensate for lower amount of loaded RNA into RT and subsequent SiMSen-Seq. Mean + 95% CI is shown, $n = 3$. **B)** Number of consensus reads for all individual *TP53* assays. Mean + 95% CI is shown, $n = 3$. **C)** Coefficient of variation among individual RT conditions calculated using consensus reads. Note that replicates were performed at RNA level when evaluating RT conditions, $n = 3$. **D)** Dynamic range of Omniscrypt. Dilution series ranging from 140 to 8.75 ng total RNA. One sample at 35 ng total RNA was considered outlier and removed (Supplemental Table 4). Linear regression was performed for each assay using the four lowest RNA concentrations to guide the eye, $n = 2 - 3$. **E)** Complementary DNA yield for hotspot mutation panel. Omniscrypt was used with 200, 5 and 1 ng total RNA. Mean + 95% CI is shown, $n = 3$.

Figure 4. Error rates in digital RNA sequencing. **A)** Mean error rates. The error rate was calculated as the total number of non-reference reads divided by the total number of detected reads per nucleotide position using consensus reads. The errors were then averaged for all nucleotide positions and assays. Mean + 95% CI is shown, $n = 3$. **B)** Error rate per individual assay. Mean + 95% CI is shown, $n = 3$. **C)** Linear relationship between total number of non-reference molecules compared with total number of detected molecules using consensus reads. The mean for each RT condition and genomic DNA is shown. The linear regression is shown. **D)** Error correction factor using UMIs. The error rates before and after using UMIs were calculated and compared. Mean + 95% CI is shown, $n = 3$. **E)** Error rates for individual assays in the hotspot mutation panel. Omniscript was used with 200 ng total RNA. Mean + 95% CI is shown, $n = 3$. **F)** Error correction factor using UMIs for hotspot mutation panel. The error rates before and after using UMIs were calculated and compared. Mean + 95% CI is shown, $n = 3$. * indicates that no factor value could be calculated since no errors were detected after UMI correction.

A)

B)

Figure 5. Types of errors in digital RNA sequencing. A) Sequence-context dependent errors. Spearman correlation coefficients for error rates between all RT conditions and genomic DNA. The error rate was calculated for each nucleotide position using data for all five *TP53* assays. Statistically significant values ($p < 0.05$) are underlined. **B)** Error rates for substitutions including transitions and transversions (gray background), deletions and insertions for all RT conditions and genomic DNA. Data are shown for raw reads and UMI-error corrected reads. Mean + 95% CI is shown, $n = 3$.

Figure 6. Detection of mutations using digital RNA sequencing. **A)** Detection of spike-in *TP53* single nucleotide variant. The observed mutant allele frequency is shown at the y-axis, while the expected spike-in mutant allele frequency is shown at the x-axis. The *TP53* assay F, analyzing 200 ng total RNA or 10 ng genomic DNA was used, $n = 3$. *** $p \leq 0.001$, unpaired Student's t-test. **B)** Mutation detection and error rates. The error rates for samples with a spike-in mutation (gray) and controls without the spike-in mutation (black) samples are shown. The spike-in mutation position is indicated by red color with an expected mutant allele frequency of 0.078%. Mean + 95% CI is shown, $n = 3$. **** $p \leq 0.0001$, two-way ANOVA with Šidák correction. **C)** Breast cancer panel cDNA yields. Relative number of consensus reads for breast cancer panel analyzing 200 ng total RNA, 20 ng matching genomic DNA as well as 20 ng reference DNA. Mean + 95% CI is shown for the left panel, $n = 3$. Mean only is shown for the right panel since no CI can be determined, $n = 3$. **D)** Detection of breast cancer mutations. The observed mutant allele frequency is shown for cDNA and genomic DNA for three breast cancer samples. Note that the variation in *AZIN1* (blue), caused by RNA editing, is only detected at RNA level. **E)** Summary of mutations called by global RNA sequencing (RNA-Seq) and SiMSen-Seq.

Figure 7. Detection of cell-free RNA in blood plasma. **A)** Number of detected cDNA molecules per mL blood plasma calculated from consensus reads. The mean of all five assays is shown to the right. Mean + 95% CI is shown, n = 3. **B)** Number of consensus reads for five *HBB* assays analyzing 10 ng genomic DNA as reference. The mean of all five assays is shown to the right. Mean + 95% CI is shown, n = 3. **C)** Error rate per assay. The mean of all five assays is shown to the right. Mean + 95% CI is shown, n = 3. **D)** Error correction factor using UMIs. The error rates before and after using UMIs were calculated and compared. Mean + 95% CI is shown, n = 3. Data for all nucleotide positions with and without UMIs are shown in Supplemental Figure 8. **E)** Number of detected cDNA molecules per mL blood plasma for hotspot mutation- and breast cancer panel, respectively. Mean + 95% CI is shown, n = 3. * indicates no detected molecules.

Supplemental figures

Supplemental Figure 1. UMI handling and bioinformatics. **A)** The concept of UMIs. After sequencing, all molecules originating from the same initial molecule (I-IV) are labeled with identical UMIs (red, green, blue and yellow). For every target sequence, reads with identical UMIs can be bioinformatically collapsed into one consensus read per UMI. To call a mutation, the majority of all reads with the same identical UMI also need to have identical allele variant. Hence, molecules with true mutations (molecule II) can be identified and technical errors (molecule III) will be corrected. The use of UMIs will also help reduce quantification biases since all reads with the same identical UMI are collapsed into a single consensus read. For example, molecule I and IV are both generating one consensus read each, despite having different number of reads per UMI. **B)** Schematic overview of bioinformatics pipeline using `UMIErrorCorrect`²⁸. This Python package consists of three subprocesses: fastq data preprocessing, mapping to reference genome and consensus reads generation.

Supplemental Figure 2. UMI tagging during barcoding PCR. A) Barcoding PCR using single-stranded complementary DNA as input. The output includes 3 uniquely barcoded molecules. **B)** Barcoding PCR using double-stranded genomic DNA as input. The output includes 6 uniquely barcoded molecules.

Supplemental Figure 3. Assay positions in *TP53* and *HBB* genes. A) Barcoding PCR using single-stranded complementary DNA as input. The output includes three uniquely barcoded molecules. B) Barcoding PCR using double-stranded genomic DNA as input. The output includes six uniquely barcoded molecules.

Supplemental Figure 4. UMI-error correction. **A)** Error rates before and after UMI-correction are shown for each nucleotide position. Mean \pm SD is shown, $n = 3$. **B)** Outlier data. Nucleotide position 7669645 in *TP53* displayed high error rates before and after UMI-error correction for a subset of RT conditions. The number of consensus reads for each nucleotide base type (A, C, G and T) is shown as well as the number of deletions (D) and insertions (I). Mean values are shown, $n = 3$.

Supplemental Figure 5. Sequencing errors in relation to sequence context. The error frequency per nucleotide position after UMI correction for all reverse transcription condition in *TP53* is shown. The outlier nucleotide position 7669645 was excluded for visualization purpose.

Supplemental Figure 6. Breast cancer panel performance. **A)** Complementary DNA yield. Number of consensus reads for *TP53* assay F, analyzing 200 ng total RNA. Ten nanograms genomic DNA was analyzed as control. The expected spike-in mutant allele frequency is shown at the x-axis. Mean \pm 95% CI is shown, $n = 3$. **B)** Mean error rates across all assays. The error rate was calculated as the total number of non-reference reads divided by the total number of detected reads per nucleotide position using consensus reads. The errors were then averaged for all nucleotide positions and assays. Mean \pm 95% CI is shown, $n = 3$. **C)** Error rate per assay. Mean \pm 95% CI is shown, $n = 3$. **D)** Error correction factor using UMIs. The error rates before and after using UMIs were calculated and compared. Mean \pm 95% CI is shown, $n = 3$. **E)** Detection of mutations in diluted RNA samples. The number of mutated *SF3B1* molecules is shown when the total RNA from patient 1 was diluted with total RNA from MLS 1765-92 cells, keeping the amount of total RNA constant at 200 ng in RT, $n = 1$. In the 1:64 dilution sample, three molecules were observed, while no molecules were detected in the 1:256 dilution sample as well as in the control sample containing only MLS 1765-92 total RNA.

Supplemental Figure 7. Detection of *TP53* mutation with and without UMI correction. The mean error rates for samples with (gray) and without (black) spike-in mutant molecules are shown, excluding the nucleotide position with mutation. Mean and 95% confidence of distribution is shown (Mean + 1.96 SD, n = 42). Mean + 95% CI is shown for the mutated nucleotide position (red), n = 3.

Supplemental Figure 8. *HBB* error rates. *HBB* data across all five assays with and without error correction using UMIs. The error rate per position was calculated as the total number of non-reference reads divided by the total number of detected reads with and without UMI-error correction. Mean + 95% CI is shown, n = 3.

REVIEWERS' COMMENTS:

Reviewer #1 (Remarks to the Author):

The authors have added some useful data to the study and should be commended for the careful preparation of the revised version. There are some additional comments:

1. For the very last part of the results, the authors added some data on the hotspot mutation and breast cancer panels on cell-free RNA from blood plasma of healthy individuals, and, unsurprisingly, found no mutations, but also there were few transcripts overall. It is quite surprising the authors did not sequence cell-free RNA from cancer patients. One would hope/expect that to work much better and that would have been a much better use case to illustrate clinical utility.
2. Lines 262: given the number of reads of FLT3, it is not fair to claim 'no detected error'. It would be sufficient to say the max mean error rate was 0.0055% for NOTCH1.
3. Lines 261-269 should probably refer to Figures 4E-F.

Reviewer #2 (Remarks to the Author):

Thanks to all the authors for answering my questions. I am satisfied with all the new information they have added to support their findings. This method will be very useful for genomics researchers especially in relation to cancer research.

Reviewer #3 (Remarks to the Author):

The authors went to great lengths in addressing the reviewers' comments and substantially improved the manuscript. I only have a few minor comments that I leave to the authors and editor to address.

1. The authors state: "...in liquid biopsies the theoretical limit of detection is 5% if we in total only can detect 20 molecules." The limit of detection is generally expressed as the analyte concentration whereby 95% of the replicates are still positive, also called LOD95. As the theoretical lowest LOD95 is 3 molecules (because of Poisson sampling), I do not entirely agree with the authors' statement. You need, thus, 3 mutant cDNA molecules in the reaction to be 95% sure to detect one. With 20 RNA molecules (not even discussing extraction, RT conversion, ...), you thus can have, at best, 15% VAF LOD95.
2. I remain skeptical of using the word 'digital'. It is not a yes/no method. Digital does not mean one by one, but either 0 or 1.
3. With other quantification biases, I was not implying that UMI could solve them. Rather a warning to the reader that there are other errors/biases, apart from sequencing errors (that are effectively dealt with using the authors' method), e.g. nucleic acid extraction bias, RT bias and artifacts, PCR bias (length, GC, structures, ...).

We thank the reviewers for their thorough work and their useful feedback, which has been taken in consideration to improve this manuscript. We addressed their final comments as outlined below and all changes have been marked in the manuscript.

Reviewer #1 (Remarks to the Author):

1. For the very last part of the results, the authors added some data on the hotspot mutation and breast cancer panels on cell-free RNA from blood plasma of healthy individuals, and, unsurprising, found no mutations, but also there were few transcripts overall. It is quite surprising the authors did not sequence cell-free RNA from cancer patients. One would hope/expect that to work much better and that would have been a much better use case to illustrate clinical utility.

Indeed, it will be interesting to study mutations in cell-free RNA from cancer patients. However, it is beyond the scope of this method-oriented study to analyze cohorts of cancer patients. This will be a highly relevant follow-up study, focusing on clinical utility.

2. Lines 262: given the number of reads of FLT3, it is not fair to claim 'no detected error'. It would be sufficient to say the max mean error rate was 0.0055% for NOTCH1.

Agreed, we have updated the text only referring to the maximum mean error rate.

3. Lines 261-269 should probably refer to Figures 4E-F.

Indeed, we have updated the text referring to Figures 4E-F.

Reviewer #3 (Remarks to the Author):

1. The authors state: "...in liquid biopsies the theoretical limit of detection is 5% if we in total only can detect 20 molecules." The limit of detection is generally expressed as the analyte concentration whereby 95% of the replicates are still positive, also called LOD95. As the theoretical lowest LOD95 is 3 molecules (because of Poisson sampling), I do not entirely agree with the authors' statement. You need, thus, 3 mutant cDNA molecules in the reaction to be 95% sure to detect one. With 20 RNA molecules (not even discussing extraction, RT conversion, ...), you thus can have, at best, 15% VAF LOD95.

Agreed, this reviewer response by us was not fully correct. We simply referred to the fact that if we detect 20 molecules, one molecule corresponds to 5% (1 out of 20), which is not that meaningful information. This does not affect the manuscript text since it was not included.

2. I remain skeptical of using the word 'digital'. It is not a yes/no method. Digital does not mean one by one, but either 0 or 1.

We are using 0 and 1 in our approach. If we detect less than 3 reads per UMI family we do not use the reads to form a consensus read (= 0), while if we have ≥ 3 reads per UMI family we form consensus reads (= 1) and data are used for downstream analysis. Hence, it is very similar to digital PCR. Digital sequencing is not a new term but has been used before. We think it is informative since we are converting sequencing reads to 0 and 1 in the preprocessing step.

3. With other quantification biases, I was not implying that UMI could solve them. Rather a warning to the reader that there are other errors/biases, apart from sequencing errors (that are effectively dealt with using the authors' method), e.g. nucleic acid extraction bias, RT bias and artifacts, PCR bias (length, GC, structures, ...).

Agreed, we apologize for miss interpretation of this important comment. These limitations of UMIs were incorporated in the last revised manuscript at several places in the discussion. For example, we included an extended discussion about errors in the RT step that cannot be handled by UMIs, but all mentioned biases are mentioned in the manuscript.